# Thermochemical oxidation of methane induced by high-valence metal oxides in a sedimentary basin

Wen-Xuan Hu[1], Xun Kang[1], Jian Cao[1], Xiao-Lin Wang[1], Bin Fu[2] & Hai-Guang Wu[1]

Thermochemical oxidation of methane (TOM) by high-valence metal oxides in geological systems and its potential role as a methane sink remain poorly understood. Here we present evidence of TOM induced by high-valence metal oxides in the Junggar Basin, located in northwestern China. During diagenesis, methane from deeper source strata is abiotically oxidized by high-valence Mn(Fe) oxides at 90 to 135 °C, releasing $^{13}C$-depleted $CO_2$, soluble $Mn^{2+}$ and $Fe^{2+}$. Mn generally plays the dominant role compared to Fe, due to its lower Gibbs free energy increment during oxidation. Both $CO_2$ and metal ions are then incorporated into authigenic calcites, which are characterized by extremely negative $\delta^{13}C$ values (−70 to −22.5‰) and high Mn content (average MnO = 5 wt.%). We estimate that as much as 1224 Tg of methane could be oxidized in the study area. TOM is unfavorable for gas accumulation but may act as a major methane sink in the deep crustal carbon cycle.

[1] State Key Laboratory for Mineral Deposits Research, School of Earth Sciences and Engineering, Nanjing University, Nanjing 210023, China. [2] Research School of Earth Science, The Australian National University, Canberra ACT 0200, Australia. These authors contributed equally: Wen-Xuan Hu, Xun Kang. Correspondence and requests for materials should be addressed to W.-X.H. (email: huwx@nju.edu.cn) or to X.K. (email: kangxunk@163.com)

Methane ($CH_4$) is an economically important fossil fuel, and by 2014 about $4.3 \times 10^{14}$ m³ or $4.7 \times 10^5$ Tg of proven natural gas reserves, including shale gas, had been identified in sedimentary basins worldwide[1,2]. However, methane is also a significant greenhouse gas, second only to $CO_2$ in the contribution to the current global greenhouse effect[3,4]; during certain intervals of the geologic past, such as the Archean, it was perhaps the most significant greenhouse gas in the atmosphere[5,6]. Since the late Holocene pre-industrial era, the concentration of atmospheric methane had increased sharply from an average of 680 p.p.b.v. (parts per billion volume) during AD 800–1600 to 1799 p.p.b.v. by the year 2010[4,7]. Furthermore, the current emission flux of methane from sedimentary basins to the atmosphere is estimated to be 10 to 25 Tg yr⁻¹, which is a key to understanding the global carbon cycle and climate change[8,9].

Oxidation reactions are the key mechanism by which methane is consumed before emission into the atmosphere. Methane can theoretically be oxidized via two distinct geological processes. The first is anaerobic oxidation of methane (AOM), where methane oxidation is mediated by sulfate- or metal-reducing bacteria in an anoxic environment, typically resulting in the precipitation of authigenic carbonate cements[10–12]. The second is thermochemical oxidation of methane (TOM) at high temperatures by sulfate or high-valence metal oxides. Like the bacterially-mediated AOM reaction, this process typically results in the formation of authigenic carbonate[13–16].

Natural gas reservoirs in sedimentary basins are typically buried to depths of several kilometers, and experience temperatures from 60 to 150 °C[17], while the metabolism of the archaeal groups responsible for AOM are mainly active at temperatures below 80 °C[18,19], limiting the depth at which archaea dominate the conversion of $CH_4$ to $CO_2$. Therefore, TOM is potentially the predominant methane oxidation process in the strata with high formation temperature (especially > 80 °C)[13–15,20].

Thermochemical sulfate reduction (TSR) consuming methane has been observed to occur in geological systems[13–15,20]. Its reaction conditions, as well as geological and geochemical responses, have been deeply explored[13–15,20,21]. In contrast, almost no geological evidence has been reported for the oxidation of methane with high-valence metal oxides, except for the oxidation of methane by $Fe^{3+}$ in detrital biotite at temperatures above 270 °C during metamorphism[16,22,23]. To date, the process has only been simulated in the laboratory setting at high temperatures (350 to 650 °C)[24–27]. Thus, both the extent of TOM by high-valence metal oxides in geological systems and its role in the methane sink remain poorly understood.

The Junggar Basin is a major petroliferous basin in northwestern China, with an area of approximately $1.3 \times 10^5$ km²[28]. The Lower Triassic Baikouquan Formation ($T_1b$) is located on the western slope of the Mahu Sag, a structure forming part of the northwestern margin of the Junggar Basin (Fig. 1a). The Mahu Sag is the richest hydrocarbon reservoir in the Junggar Basin, with approximately two billion tons of proven crude oil reserves[29]. Continuous oil and gas reservoirs were recently discovered in the Baikouquan Formation (Fig. 1a), with the hydrocarbons believed to be derived largely from the dark, organic-rich mudstones of the underlying Lower Permian Fengcheng Formation (see Supplementary Note 1 and Supplementary Fig. 1 for detailed descriptions)[28,30].

The Baikouquan Formation can be divided into three members, referred to as $T_1b_1$, $T_1b_2$, and $T_1b_3$ in ascending stratigraphic order. The maximum burial depth of the unit from west to east across the Mahu Sag increases from 2850 m to 4300 m. The formation consists of interlayered mudstone and sandy conglomerate, deposited in a succession of alluvial fan-deltaic systems with abundant gravity flow deposits[31,32]. The brown or greyish-green conglomeratic layers contain pebble conglomerates, sandy gravel conglomerates, and sandstones, whereas the mudstone layers are dominated by massive brown silty mudstones with minor gray laminations (Fig. 1b). The abundance of brown mudstone increases up-section in three members, which is interpreted to reflect an arid-semiarid climate regime and oxic lacustrine environment during deposition[31,33].

The $T_1b$ sediments are derived from two provenances, i.e., the basement, comprising granite and mafic-ultramafic igneous rocks, and the sedimentary strata from the Carboniferous to the Permian[28,31]. The weathering of mafic-ultramafic rocks and tuffaceous components in sedimentary rocks forms abundant high-valence Fe-Mn oxides during $T_1b$ deposition interval (see Supplementary Note 2 for provenance analysis)[34].

In this study, we report the first geological evidence for extensive TOM induced by high-valence metal oxides in the Lower Triassic reservoir strata of the Junggar Basin. We find that methane was abiotically oxidized by high-valence Mn(Fe) oxides at high temperatures (90–135 °C) in the deep burial strata, ultimately yielding extremely $^{13}C$-depleted authigenic calcites with high manganese contents. In the process thousands of Tg of $CH_4$ were consumed, implying TOM may act as a major $CH_4$ sink in the deep carbon cycles.

## Results

**Distribution and geochemistry of calcite cements**. Authigenic calcite in the sandy conglomerates of the Baikouquan Formation occurs mainly as coarsely crystalline cement (Fig. 2a, b), sometimes within feldspar along cleavage (Fig. 2a). Based on microscopic observations, the total content of calcite cement in the sandy conglomerates ranges from 0 to 6% by area, with an average of ~2.5%. There are two stages of precipitation: the early-stage calcite occurs as xenomorphic crystals filling dissolved pores or poorly-connected interparticle pores (Fig. 2a), whereas the late-stage cements appear as coarse crystals, with well-developed crystal planes in the central portions of large interparticle pores (Fig. 2a, b).

Electron probe micro-analysis (EPMA) reveals that these authigenic calcites are enriched in manganese, with MnO content ranging from 0.79 to 14.67 wt.% (average = 5.05 wt.%; Fig. 3a). In detail, the MnO content of early-stage calcite cements is generally lower than 4.00 wt.%, while late-stage cements typically have MnO content greater than 5.00 wt.%, some up to 11.00–15.00 wt.% (Fig. 3a). In contrast, the FeO content of the calcite cements is very low, ranging from<0.01 to 0.79 wt.%, with an average of 0.12 wt.% (Fig. 3b).

Authigenic calcite cements in the $T_1b$ reservoir rocks are very negative in $\delta^{13}C$. Whole-rock carbonate $\delta^{13}C$ values range from −69.8 to −22.5‰, though most samples fall within the −50.0 to −36.0‰ range (Fig. 4a). The calcite cements have $\delta^{18}O$ values ranging from −22.6 to −12.8‰ (Fig. 4b).

In situ carbon isotope analyses reveal that the two generations of calcite cements have markedly different $\delta^{13}C$ values, sometimes by 10‰ (Fig. 2c, d). In contrast, the difference in $\delta^{18}O$ between the two generations of cement is relatively small, with a maximum offset of 1.9‰ (Supplementary Table 1). To better constrain the sources of the carbon and oxygen incorporated into calcite cements in the $T_1b$ reservoir, we also measured the isotopic composition of calcite in the underlying source rocks: −22.4 to −13.5‰ for $\delta^{13}C$; and −18.5 to −12.8‰ for $\delta^{18}O$.

**Geochemistry of hydrocarbons**. In most natural gas samples more than 95 mol. % hydrocarbons are light components, i.e., $C_1$–$C_7$. $CH_4$ is the most abundant hydrocarbon gas, with an aridity coefficient of 0.77–0.95. The non-hydrocarbon component

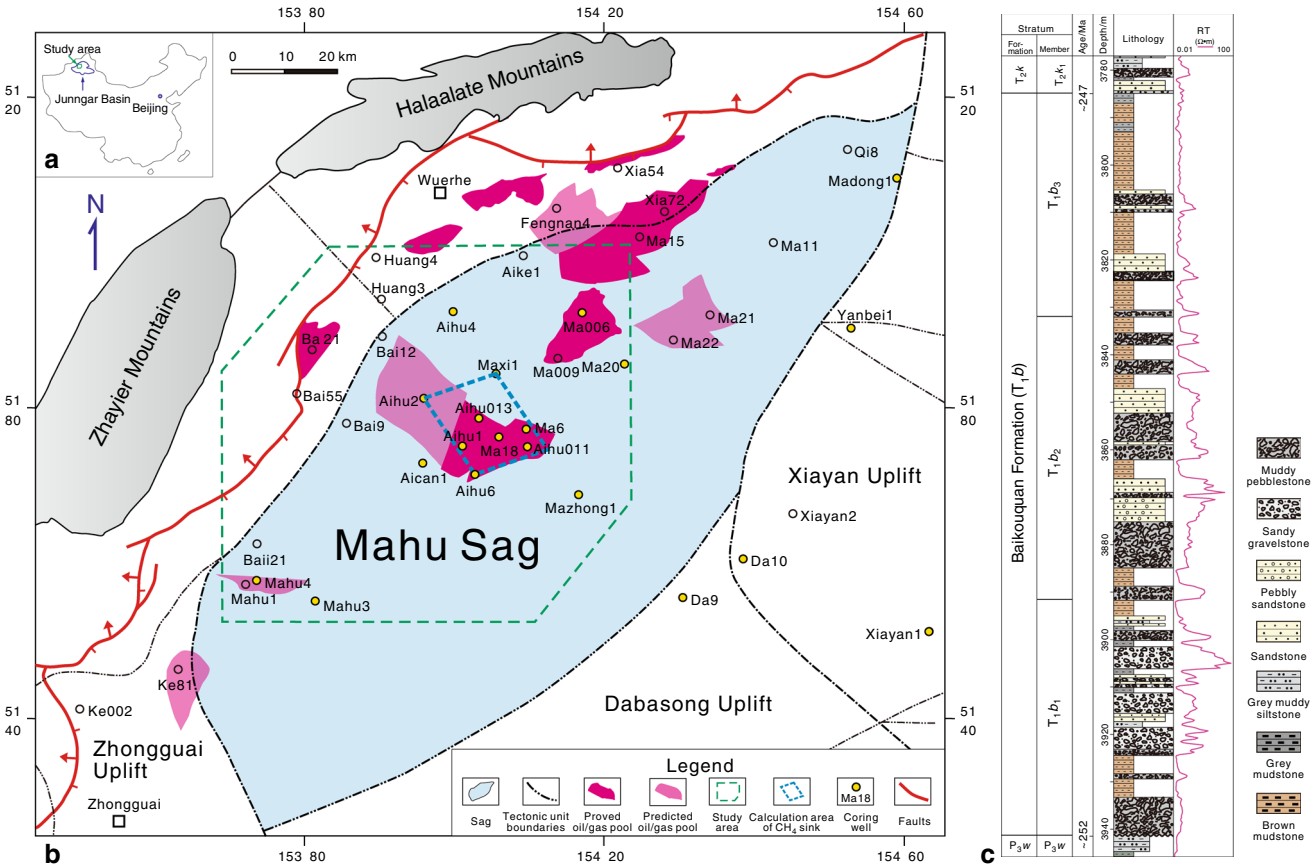

**Fig. 1** Schematic geologic map of the Mahu Sag, and generalized stratigraphy of the Lower Triassic Baikouquan Formation. **a** The location of the Junggar Basin in China. **b** The study area is located in the west slope of the Mahu Sag adjacent to the northwestern marginal thrust faults of the Junggar Basin. **c** The sun-shape symbols within the Resistivity logging column indicate reservoir beds rich oil and gas

consists mainly of $N_2$, with minor $CO_2$ and no detectable $H_2S$. The ranges of $\delta^{13}C$ values in methane, ethane, propane, and butane are −39.3 to −46.8‰, −27.5 to −31.9‰, −27.2 to −30.2‰, and −26.0 to −28.8‰, respectively (Supplementary Table 2).

Gas chromatography of crude oil samples from $T_1b$ reveals a complete sequence of n-alkanes (Fig. 5), especially ~$C_8$–$C_{12}$ which are preferentially removed in the earliest stages of biodegradation[19]. This indicates that the hydrocarbons have not experienced microbial degradation. The absence of 25-norhopanes and other representative values of key parameters also show no evidence for biodegradation (Supplementary Table 3). Biomarker analyses are described in details in the Methods section.

## Discussion

The extremely negative $\delta^{13}C$ values of authigenic calcite can be interpreted to be derived from organic matter[35,36]. In organic matter-enriched geological systems, both oxidation of organic matter and thermal decarboxylation of organic acids can generate significantly $^{13}C$-depleted $CO_2$, which may then be incorporated into authigenic calcite[11–13,27,35]. The $\delta^{13}C$ value of authigenic calcite formed via the oxidation of hydrocarbons can be as low as −125‰[11–13,36], whereas calcite precipitated from the decarboxylation of organic acids concentrates from −3 to −25‰, rarely yielding values below −25‰[35,37]. Authigenic calcite in the study area shows $\delta^{13}C$ values falling mainly into the range of −50.0 to −36.0‰, with a minimum value of −69.8‰, consistent with the oxidation model for gaseous hydrocarbons[11,12,36].

During the oxidation of gaseous hydrocarbons, $^{12}C$ is preferentially oxidized to $CO_2$, leaving the remaining hydrocarbon pool to become increasingly enriched in $^{13}C$ as the reaction proceeds[24,38,39]. Figure 6 shows that the $\delta^{13}C$ value of methane in $T_1b$ natural gas samples increases from around −45‰ to values above −40‰ as the $CO_2$ contents of the gas increases; however, the isotopic composition of heavier hydrocarbon gases ($C_2$–$C_4$) remains comparatively stable. This result demonstrates that, within the $T_1b$ natural gas pool, abundant $CH_4$ was oxidized during diagenesis. The $\delta^{13}C$ values of the authigenic calcite in $T_1b$ are generally lower than those of the associated methane (−46.8 to 39.3‰; Fig. 7), further supporting our interpretation that the calcite is mainly derived from $CO_2$ generated by methane oxidation[11,12,36].

There are two mechanisms for oxidization of methane in sedimentary systems, i.e., AOM and TOM[10,11,13–15]. AOM has been observed mainly in modern marine sediments, especially near cold seeps on the seafloor, where the $\delta^{13}C$ values of the associated authigenic carbonate can be as low as −125‰[10,11,36]. TOM can occur with or without TSR reactions. Methane is usually not the most active reductant in systems where TSR occurs compared with $C_{2+}$ gaseous alkanes, so the minimum $\delta^{13}C$ value of the associated calcite is not usually lower than −31‰[13,15,39,40]. TOM reactions induced by high-valence metal oxides instead of sulfate have never been reported in natural systems, though they have been simulated in the laboratory[24–27].

All the three processes described above may have contributed to the precipitation of calcites in $T_1b$. The range of measured $\delta^{13}C$ values (−69.8 to −22.5‰) is compatible with calcite formed via AOM, but is also lower than the typical value of calcite formed via

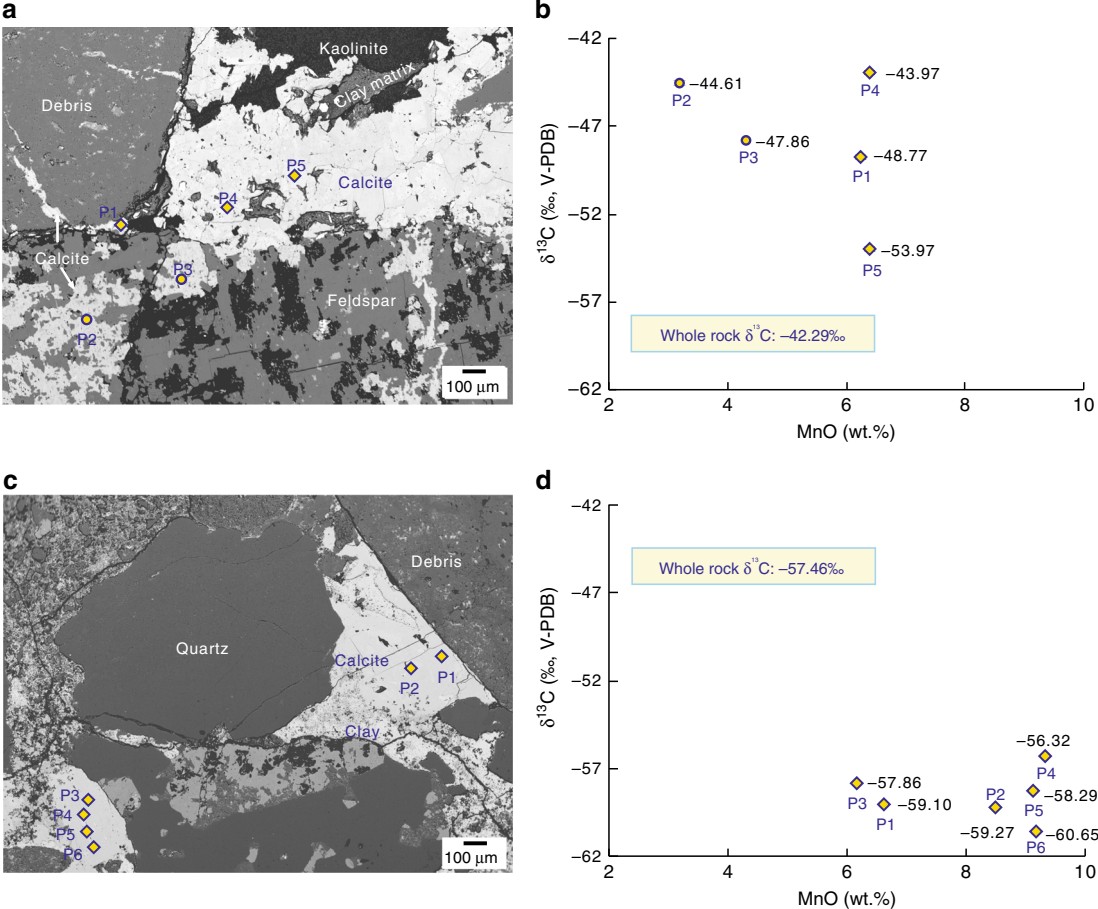

**Fig. 2** Back scattered electron microscopic images of sandy conglomerates from the Baikouquan Formation, with crossplots of in situ $\delta^{13}C$ and Mn contents of authigenic calcite. **a**, **c** Sandy conglomerate from the $T_1b_3$ member (3824.2 m in Well M18), with two generations of calcite cement. The early-stage cements have consistently lower MnO content than the late-stage cements. **b**, **d** Sandy conglomerate from the $T_1b_2$ member (3866.9 m in Well M18), cemented by calcite with higher MnO content. Circles indicate measurements of early-stage calcite, diamonds represent measurements of late-stage cements. The overall values of late-stage calcite $\delta^{13}C$ are lower than those of the early-stage calcite. The error bars, referring to the standard error of the mean (SEM) in this paper, is smaller than the symbol size in the cross-plots

methane-dominated TSR ($\delta^{13}C = -31‰$)[13]. While there are no carbon isotope data reported for the carbonate associated with TOM induced by high-valence metal oxides in geological systems, the fractionation factors may be assumed to be similar to those in laboratory-simulated experiments. Thus, we cannot rule out the TOM process based on our isotopic measurements.

Anaerobic oxidation of methane by bacteria or archaea can be ruled out. Firstly, although archaeal groups such as ANME-1 and ANME-2 can anaerobically metabolize methane in the presence of oxidants such as $MnO_2$, $SO_4^{2-}$, and $Fe^{3+}$ at temperatures $< 80\,°C$[10,18,19,41], the high formation temperature ($> 80\,°C$) does not favor the metabolism of these archaeal groups[19]. The $T_1b$ reservoir strata were buried at depths of 2250–3200 m when they were firstly charged with methane in the Early Jurassic[32]. Taking the geothermal gradient as $28–32\,°C\,km^{-1}$ and the surface temperature as $20\,°C$ during the Early Jurassic[42], this corresponds to a formation temperature of $83–122\,°C$. The $\delta^{18}O$ values of authigenic calcites also indicate precipitation of the cements at relatively high temperatures. A similar diagenetic condition was reported for the Brent Group of the North Sea with a burial depth of 2500–3500 m[43,44]. The $\delta^{18}O$ value of $T_1b$ pore water is likely to have increased by the same value (about 4‰) as the Brent Group from that of the initial meteoric water ($-10$ to $-8‰$) to $-6$ to $-4‰$[44–46]. Using this $\delta^{18}O$ value, we calculated that the temperature was greater than 90 °C during formation of more than

85% of the total calcite, with a background temperature range of 90–135 °C and an average of 109.6 °C (see Supplementary Data 2), consistent with the temperatures estimated from the geothermal gradient.

Secondly, $T_1b$ hydrocarbons show no evidence of biological degradation, indicating that archaea had negligible influence on the composition of oil and gas. The first manifestation of hydrocarbon biodegradation is typically the selective removal of $C_8–C_{12}$ normal alkanes[19]. Light $C_2–C_6$ hydrocarbons exhibit a consistent degradation sequence: propane is the first compound to be altered, followed by butane, pentane, and $C_{6+}$ hydrocarbons[47]. The net effect of biodegradation is an overall decrease in wet gas components ($C_{2+}$) and enrichment in methane[19]. The crude oil hosted in the Baikouquan Formation shows a complete alkane series, with no obvious loss of $C_8–C_{12}$ alkanes, and no hump of unresolved complex mixtures (Fig. 5). Natural gas samples contain 5.14–22.68% wet gas ($C_{2+}$) and 0.79–5.23% propane. Because AOM depends on the presence of biological mediators, and there is no evidence for biological modification, we can rule out this process of methane oxidation[19,47].

Thermochemical sulfate reduction can also be ruled out. There is very little $SO_4^{2-}$ in $T_1b$ formation waters[48], and no $H_2S$ has been detected in $T_1b$ natural gas samples (Supplementary Table 2) or the associated crude oil in adjacent formations[28,29]. Pyrite is also absent in $T_1b$ reservoir strata. The lack of both key

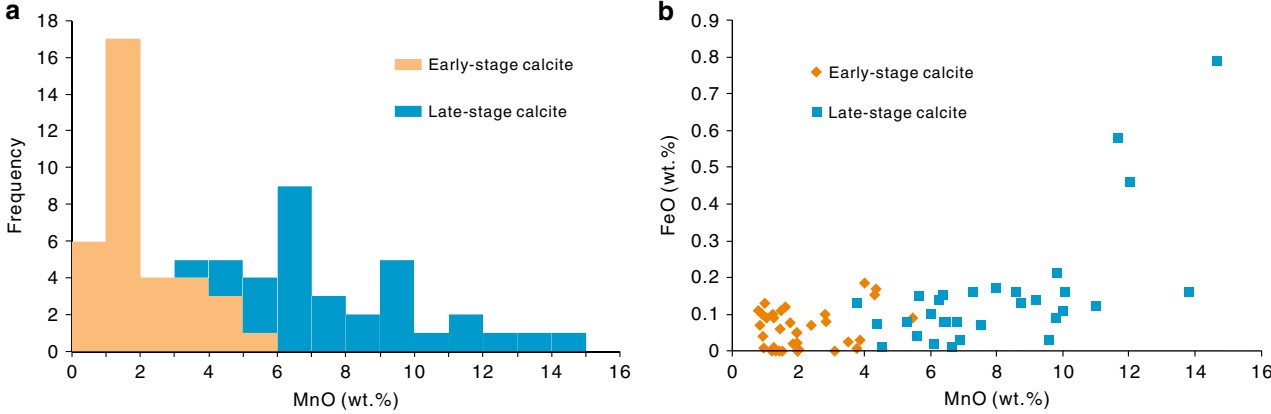

**Fig. 3** Manganese and iron contents in authigenic calcites. **a** Histogram of MnO content in early-stage and late-stage calcite cements from $T_1b$ sandy conglomerates. **b** Crossplot of MnO and FeO content in early-stage and late-stage calcite cements from $T_1b$ sandy conglomerates. The error bar (±SEM) is smaller than the symbol size. Source data are provided as Supplementary Data 1

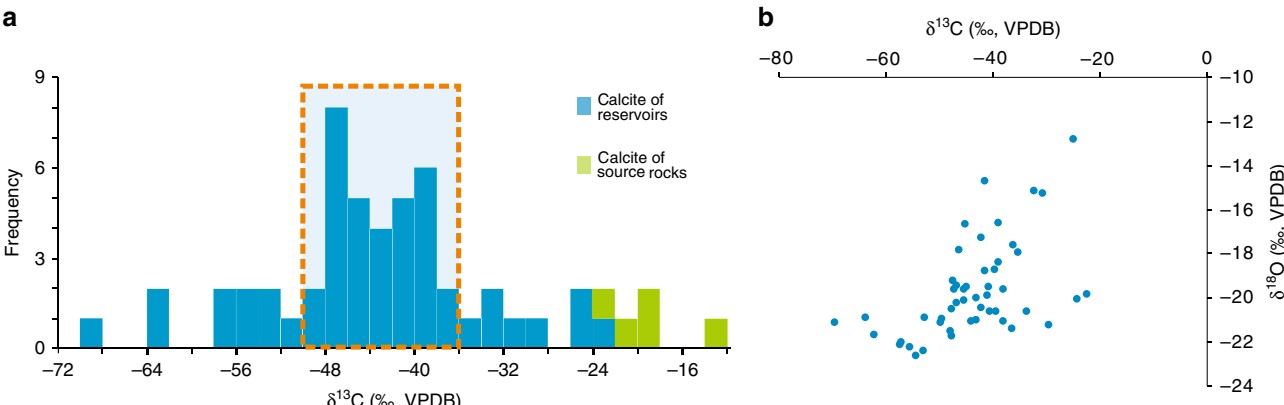

**Fig. 4** Carbon and oxygen isotopic compositions in authigenic calcites. **a** Histogram of $\delta^{13}C$ values, of calcite cements in $T_1b$ reservoirs, and calcite veins in $P_2w$ source rocks. **b** Crossplot of bulk rock $\delta^{13}C$ and $\delta^{18}O$ values in $T_1b$ calcite cements. The error bar (±SEM) is smaller than the symbol size. Source data can be seen in Supplementary Data 2

reactants (i.e., sulfate) and characteristic products (sulfide in gaseous or mineral forms) strongly suggests that the TSR reaction did not occur in our study area. Thus, the highly negative $\delta^{13}C$ values seen in the authigenic calcites cannot be attributed to methane oxidation by sulfate.

Therefore, we infer that the extremely negative $\delta^{13}C$ observed in the study area results from methane oxidation by non-sulfate oxidants at high temperatures. It is shown that methane can serve as an electron acceptor for high-valence Mn and Fe oxides, leading to the formation of $CO_2$ and the precipitation of calcite[10]. This process can be expressed with the following chemical formulas:

$$CH_4 + 4Mn_2O_3 + 15H^+ \rightarrow HCO_3^- + 8Mn^{2+} + 9H_2O$$
$$\Delta G = -442\,kJ/mol \qquad (1)$$

$$CH_4 + 4MnO_2 + 7H^+ \rightarrow HCO_3^- + 4Mn^{2+} + 5H_2O$$
$$\Delta G = -556\,kJ/mol \qquad (2)$$

$$CH_4 + 8Fe(OH)_3 + 15H^+ \rightarrow HCO_3^- + 8Fe^{2+} + 21H_2O$$
$$\Delta G = -270.3\,kJ/mol \qquad (3)$$

$$Ca^{2+} + HCO_3^- + OH^- \rightarrow CaCO_3 \downarrow + H_2O \qquad (4)$$

The Gibbs free energy increment ($\Delta G$) differs substantially between the first two reactions and the third one under the same conditions, with oxidation by $Mn_2O_3$ or $MnO_2$ being approximately twice as energetically favorable as oxidation by $Fe(OH)_3$[10]. In geological systems where high-valence Mn and Fe coexist, $CH_4$ can be preferentially oxidized by Mn rather than Fe. During the interval of $T_1b$ deposition, the northwestern margin of the Junggar Basin experienced an arid or semiarid climate[31,33]. Mafic-ultramafic rocks were weathered extensively from the provenance during the period of deposition and were ultimately deposited in the organic matter-depleted red layers of $T_1b$ (Supplementary Fig. 2). These red beds provided the material basis for subsequent methane oxidation by the high-valence Mn and Fe oxides[49]. XRF analysis shows that the Mn content of sandy conglomerates in $T_1b$ ranges from 0.07 to 0.48 wt.% (as $Mn_2O_3$) in bulk rock, with an average of 0.17 wt.%; the Fe content ranges from 2.88 to 12.71 wt.% (as $Fe_2O_3$), with an average of 6.25 wt.% (Supplementary Table 4). Further XRD, FE-SEM, and EPMA analyses indicate that high-valence Mn(Fe) exists as hematite in the clastic rocks in $T_1b$. Most of the hematite is disseminated in the clay-rich matrix, while minor occurrences as isolated amorphous aggregates are observed. The manganese generally occurs as the isomorphous $Mn^{3+/4+}$ substitution in the hematite lattice structure[50,51] at levels of 0.95–1.47 wt.% in the isolated hematite and 0.61–0.74 wt.% in the disseminated

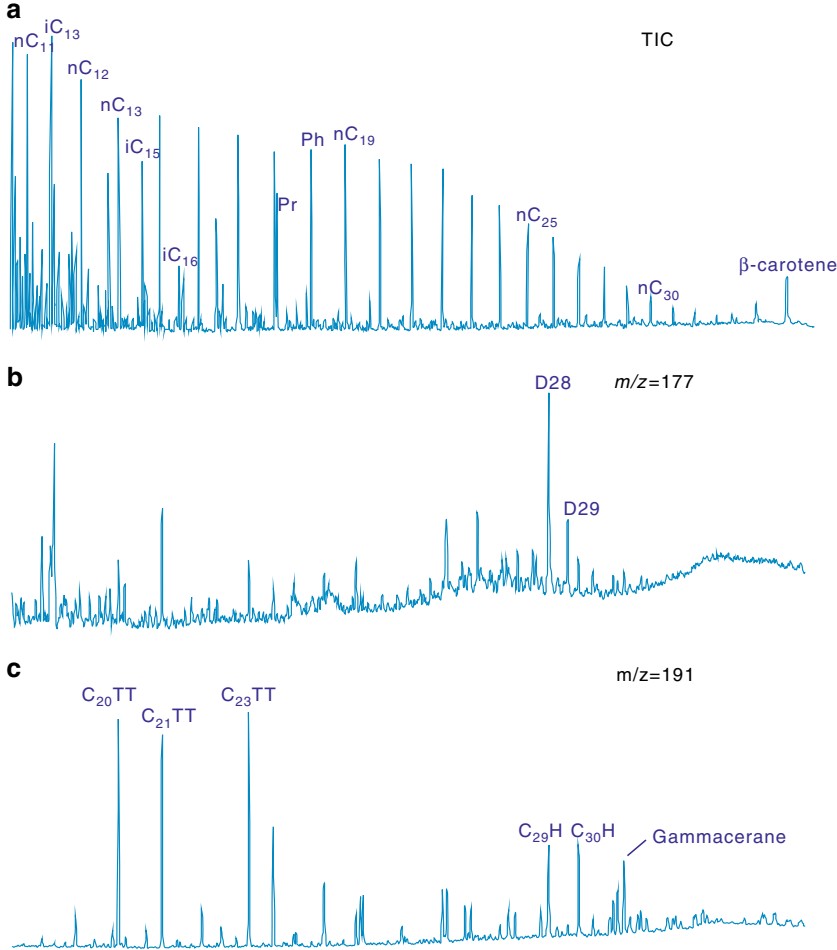

**Fig. 5** Chromatogram and mass spectrogram of a representative $T_1b$ crude oil sample (No. M18-O-1). **a** In the total ion chromatogram (TIC) Pr and Ph indicate pristane and phytane, respectively. **b** D represents demethylated hopanes in the $m/z$ 177 mass spectrogram. **c** TT is tricyclic terpane and H is hopane in the $m/z$ 191 mass spectrogram

hematite in the form of $Mn_2O_3$ (see Supplementary Note 3 and Supplementary Fig. 3 for detailed descriptions).

The authigenic calcites in $T_1b$ are generally enriched in manganese relative to whole rock samples, with MnO content ranging from 0.79 to 14.67 wt.% (average = 5.05 wt.%). However, the FeO content of the calcite cements is uniformly less than 1 wt.%, averaging at 0.12 wt.%. This reflects the energetic favorability of Mn as an oxidant relative to Fe. Thus, the precipitation of calcite was closely associated with methane oxidation by high-valence manganese.

It should be noted that $C_{2+}$ gaseous alkanes (especially ethane and propane) are usually preferentially oxidized during TSR reactions, whereas methane is the last to react[14,15,21,40]. The oxidation of abundant methane in $T_1b$ rocks has been influenced by several factors. One is the sufficient supply of $Mn^{3+/4+}$, which is present as hematite within brown clastic rocks and guarantees that almost all of the gaseous alkanes, including methane, will be consumed in the bleaching pathways[49]. The other is that since methane is the main component of the $T_1b$ natural gas, ultimately it will have dominated the thermochemical oxidation process. Moreover, the $T_1b$ clastic strata are low-permeability due to their high clay content and complex grain composition[32]. Consequently, gaseous alkanes mainly migrated by diffusion. Due to its higher initial concentration and higher diffusion coefficient, the quantity of methane reaching the matrix of reservoir rocks or mudstones at a distance from the oil-gas interface is much higher than that of $C_{2+}$ gaseous alkanes[52]. Ultimately, as shown in Fig. 6, abundant methane was oxidized in the $T_1b$ clastic rocks.

Microbial metabolism and high temperature can both serve to facilitate reactions (1) and (2). The former may reduce the energy threshold, while the latter can provide the activation energy necessary for the reaction to occur[10,24]. Since microbial activity in the $T_1b$ reservoir strata appears to have been minimal, heating would have been the only way to promote oxidation reactions. Considering the formation temperature (90–122 °C) during infiltration of natural gas and the main precipitation temperature (90–135 °C) of authigenic calcites, we can infer that methane oxidation in the study area mainly took place at 90–135 °C. Laboratory experiments have shown that reactions (1) and (2) do not occur at temperatures below 300 °C and 250 °C, respectively[24,27]. However, during extended geological processes, long intervals of time can compensate for the slow rate of reactions at low temperatures. In the case of thermochemical sulfate reduction, experiments show that the reaction occurs at temperatures greater than 250 °C[53], though evidence for TSR has been observed in sedimentary basins at temperatures of ~100 °C[40]. As another example, while the standard Rock Eval pyrolysis procedure shows the peak of hydrocarbon generation at 420–480 °C, in natural geological systems the peak of hydrocarbon generation typically occurs at 120–150 °C[54]. This also appears to be the case for TOM.

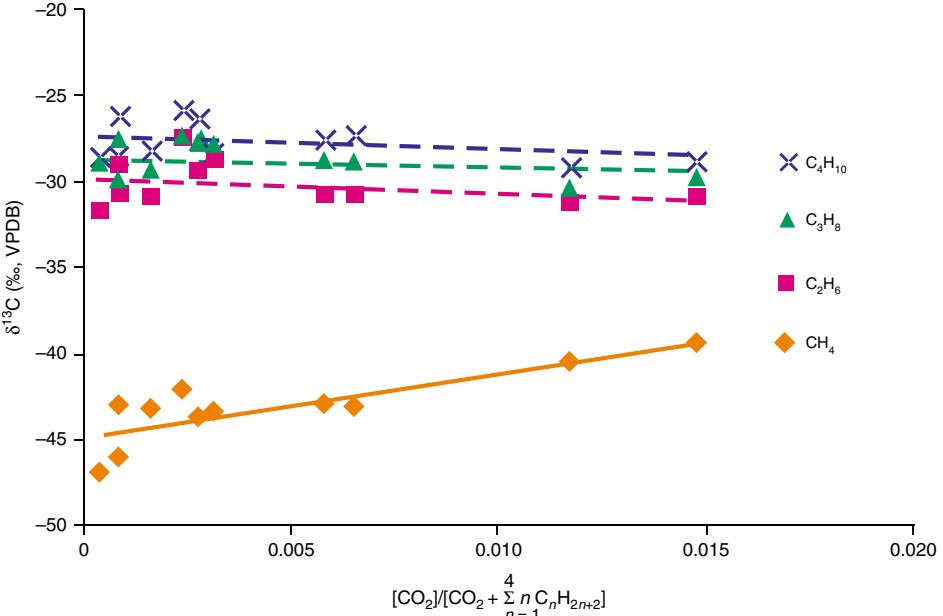

**Fig. 6** Carbon isotopes of natural gases from $T_1b$ sandy conglomerate reservoirs plotted against $[CO_2]/[CO_2 + \sum_{n=1}^{4} nC_nH_{2n+2}]$. The latter is a parameter estimating the extent of thermochemical sulfate reduction[39]. The error bar (±SEM) is smaller than the symbol size

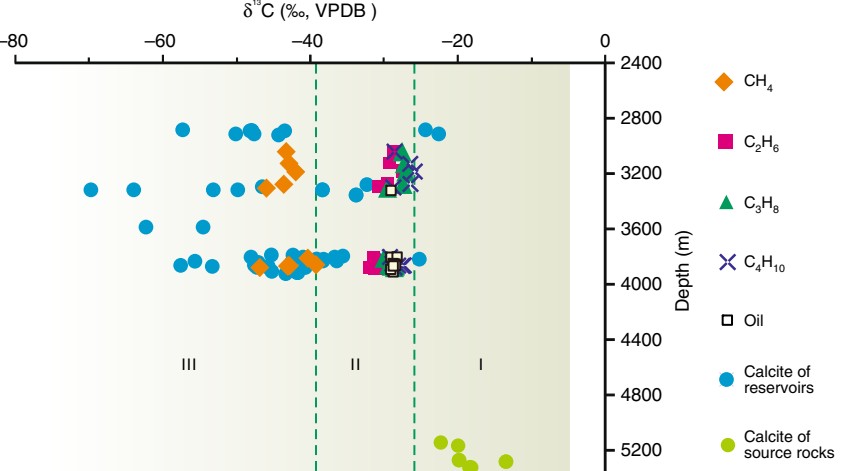

**Fig. 7** Carbon isotopes of natural gas, calcites from reservoir strata, and calcite veins from source rocks. Region I represents carbonates associated with decarboxylation of organic acids, while Region III denotes carbonates derived from the oxidation of methane. Region II indicates carbonates associated with a combination of the oxidation of methane and small quantities of $C_{2+}$, as well as decarboxylation. The error bar (±SEM) is smaller than the symbol size

The long duration of geological time can compensate for a low reaction rate at relatively low temperatures. With the increasing burial depth of $T_1b$, the formation temperature generally increased and reached 90 °C in the Middle Jurassic (~170 Ma, see Supplementary Note 4 and Supplementary Fig. 4 for thermal evolution). Since then, methane has been oxidized more efficiently.

Carbon isotope fractionation is expected to occur during thermal oxidation of methane, with the fractionation factor controlled by a number of factors, including the type of oxidant and the temperature[24,27,53]. In general, the residual methane is enriched in $^{13}C$ as the reaction proceeds, while the generated $CO_2$ is depleted, resulting in $CO_2$ with a lower $\delta^{13}C$ value than the

initial methane pool ($\delta^{13}C_{i-CH4}$). Taking the carbon isotope kinetic fractionation factor ($\alpha$) to be $10^3/(10^3-\Delta)$ in reactions (1) and (2), where $\Delta = \delta^{13}C_{i-CH4} - \delta^{13}C_{CO2}$, we calculate an $\alpha$ value of 1.0175–1.0193 for Reaction (2) at 90–135 °C[24]. This value is close to that reported for the methane-dominated TSR in the East Sichuan Basin of China (1.0166) and the ethane-dominated TSR in the Jurassic Norphlet Formation of Mobile Bay (1.109)[55]. Then, a $\Delta$ value of 17–19‰ for a closed system at 90–135 °C can be obtained. Carbon isotope fractionation also occurs during the calcite precipitation. At 90 to 135 °C, the $\delta^{13}C$ of precipitated $CaCO_3$ is expected to be 2 to 5‰ lower than that of the original gaseous $CO_2$[56]. Integrating the methane oxidation and calcite precipitation processes, the $\delta^{13}C$ of authigenic calcite is expected

to be 19 to 24‰ lower than that of the residual methane. Thus, the expected $\delta^{13}C$ of authigenic calcite is $-70$ to $-59‰$. The extremely $^{13}C$-depleted calcites may also precipitate during the oxidation of primary microbial or kerogen-cracking-derived low $\delta^{13}C$ methane, although their relative quantities involved in the reaction remain ambiguous. The former is formed via bacterial methanogenesis at temperatures generally below 50–60 °C in the $P_1f$ source rock[57], whereas the latter is generated as a concomitant of liquid oil expelled from low-maturity source rock[19]. In addition, it should be noted that the contributions of oxidation of minor $C_{2+}$ and decarboxylation of organic acids from the source rock cannot be ruled out (Fig. 8). As shown in Fig. 7, the decarboxylation reaction can produce $CO_2$ with a $\delta^{13}C$ value of around $-25‰$. The addition of small amounts of $CO_2$ from $C_{2+}$ oxidation and decarboxylation yields a range of potential $\delta^{13}C$ values in authigenic calcite from $-70$ to $-22.5‰$, consistent with our observations (Fig. 8).

Methane is generally believed to be stable in the crust, especially in sedimentary basins; oxidation is thought to occur mainly after the methane has reached the surface and been exposed to atmospheric oxygen[8,9]. The potentially important role of TOM as a methane sink in the deep strata of sedimentary basins has largely been neglected so far; methane oxidization in the deep crust has not been documented, and only a few relevant simulation experiments have been conducted to date[24,27,53]. In this study, we provide the first evidence for the occurrence of TOM in a clastic reservoir, and propose that TOM is an important $CH_4$ sink in the deep crustal carbon cycle.

In order to evaluate the extent to which TOM serves as a $CH_4$ sink, we made a series of semi-quantitative estimates based on the M18-Aihu2 oil producing area (highlighted in yellow in Fig. 1a), which contains a relatively dense concentration of exploration wells. Details of the calculations are described in the Methods section. Assuming an average authigenic calcite content of 2.5% in the relevant reservoir beds, we calculated a total authigenic calcite mass of 420 Tg. The calcite would require 57 Tg of $CH_4$ to have been oxidized, assuming only 85% of the carbonate carbon is derived from methane oxidation. If only taking account of the total oil-charging area in the sag, a fairly conservative $CH_4$ consumption is 155 Tg. However, authigenic calcites are not restricted in the conglomerates and sandstones charged by liquid oil, they also occur in the wells far away from the oil-charging area. Extending these calculations to encompass the entire Mahu Sag (~5000 km²), we estimate that as much as 1224 Tg of $CH_4$ might be oxidized in the reservoir strata. The global submarine $CH_4$ flux to the atmosphere is only 8 to 65 Tg per year[58], suggesting that TOM in sedimentary basins may play a geologically significant role in reducing $CH_4$ seepage to the surface. In other words, extensive TOM may be an important methane sink in sedimentary basins, while it is unfavorable for the accumulation of recoverable natural gas reserves.

This study reports geological evidence for widespread thermochemical oxidation of methane (TOM) induced by high-valence metal oxides in a natural environment, from the Junggar Basin in northwestern China. Methane was oxidized at high temperatures (90–135 °C) in the $T_1b$ stratum, with high-valence metal oxides acting as the electron acceptor. This reaction resulted in the precipitation of authigenic calcite cements, characterized by extremely negative $\delta^{13}C$ values ($-70$ to $-22.5‰$) and high manganese contents (average MnO = 5 wt.%). While both Mn and Fe oxides served as methane oxidants, less Fe oxide was consumed due to the relatively higher Gibbs free energy increment of the reaction. Based on the abundance and distribution of authigenic calcite, we estimate that thousands of Tg of $CH_4$ were consumed by TOM in the Lower Triassic reservoir strata of the Mahu Sag. We further propose that the geologic importance of TOM has been underestimated to date, and that this critical methane sink should be a focus of future studies of the deep carbon cycle.

## Methods

**Sampling**. A total of 103 rock samples were collected from $T_1b$ reservoir beds in 11 wells from the western slope area of the Mahu Sag (Supplementary Table 5). In addition to rock samples, 9 crude oil and 12 natural gas samples were collected from the wells (Supplementary Table 6).

**Elemental analyses**. Backscattering electron (BSE) microprobe analyses were conducted on 38 sandy conglomerate samples, using a Japan Electron Optics Laboratory (JEOL) x-ray analyzer (JXA)-8800 electron microprobe operated at 15 kV, with a 10 nA beam current. The calcite cements in 14 sandy conglomerate samples were analyzed for their major element composition, using a 2 μm beam diameter and 10 s counting time. The major element compositions of whole-rock samples were measured on a Thermo Scientific ARL 9900 × -ray fluorescence (XRF) spectrometer. The relative standard deviation of measured major element

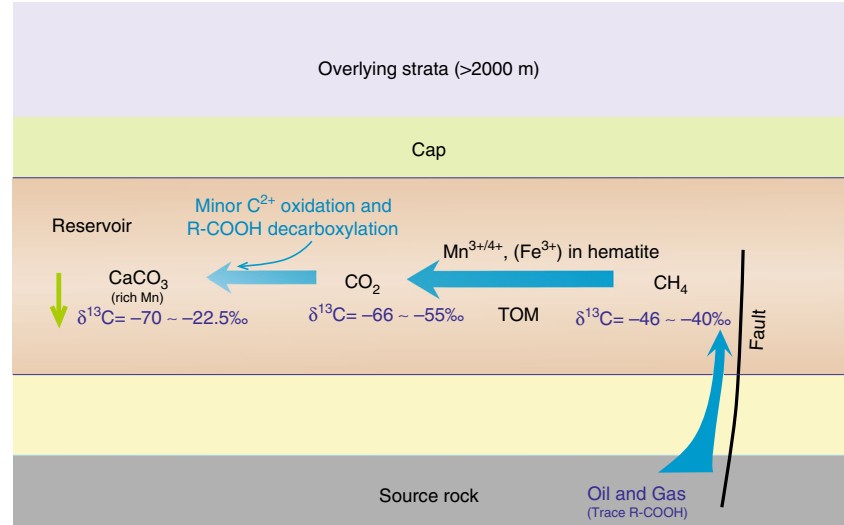

**Fig. 8** Schematic model of thermochemical methane oxidation in the clastic strata of petroliferous basins. R-COOH represents carboxylic acids. High-valence Mn (Fe) exists in hematite, which is generally found in the matrix of sandstone and conglomerate. Due to the high diffusivity of methane, minor high-valence Mn (Fe) in the brown-red mudstone adjacent to the reservoir rocks may also be reduced

concentrations is around ± 1% for elements with concentrations > 1.0 wt.%, and about ± 10% for the elements with concentrations < 1.0 wt.%, based on repeated measurements of Chinese national standards GBW-07103 (granite) and GBW-07105 (basalt). Both EPMA and XRF analyses were carried out at the State Key Laboratory for Mineral Deposits Research, Nanjing University.

**Carbonate isotope analyses of bulk rock samples.** The calcite cements in 50 $T_1b$ sandy conglomerate (i.e., reservoir rock) samples and 5 $P_2w$ mudstone (i.e., source rock) samples were analyzed for stable carbon and oxygen isotopes. Prior to analysis, each sample was broken into small pieces, and ~ 1 g of calcite cement was manually selected under a binocular microscope. All samples were then crushed to a fine powder. Twenty milligrams of each powdered sample was reacted with 100% $H_3PO_4$ at 25 °C for more than 12 h, in a Kiel IV autosampler (ThermoFinnigan). The produced $CO_2$ was then introduced into a Finnegan MAT 253 mass spectrometer for isotopic analysis. Carbon and oxygen isotope measurements were corrected using Chinese national standard GBW 04405 ($\delta^{13}C = 0.57 \pm 0.03$‰ VPDB; $\delta^{18}O = -8.49 \pm 0.13$‰ VPDB), an Ordovician carbonate from a site near Beijing. All analyses were performed at the Nanjing Institute of Geology and Paleontology, Chinese Academy of Sciences. The external precisions (1σ) of oxygen and carbon isotope analyses were ± 0.08 and ± 0.03‰, respectively.

**SIMS $\delta^{13}C_{calcite}$ analyses.** Small pieces of polished thin sections of carbonate samples were mounted in Buehler EpoFix epoxy resin, together with carbonate reference materials: NBS-19 calcite, UWC-3 calcite ($\delta^{13}C_{VPDB} = -0.91 \pm 0.08$‰, 2σ, $n = 9$; cf. ref. [59].), UW6220 dolomite ($\delta^{13}C_{VPDB} = 0.84 \pm 0.02$‰, 2σ, $n = 5$; cf. ref. [60].) and Hammerfall dolomite ($\delta^{13}C_{VPDB} = -0.28 \pm 0.07$‰, 2σ, $n = 3$; Ian S. Williams, personal communication). Backscattered electron images were taken using a JEOL JSM-6610A Analytical Scanning Electron Microscope in Research School of Earth Science (RSES), Australian National University (ANU), Canberra.

Secondary ion mass spectroscopy (SIMS) analyses of carbon isotopes were performed in situ using the Sensitive High Resolution Ion Microprobe for Stable Isotopes (SHRIMP SI) located in RSES, ANU. The analytical procedure was similar to ref. [61] for oxygen isotopes, though without electron-induced secondary ionization emission (EISIE) correction at the presence of the electron gun. Carbon isotopes ($^{12}C$ and $^{13}C$) were measured simultaneously by two Faraday cups, using an ~15 kV Cs + primary ion beam focused over an area of ~30 μm in diameter and source slit of 60 μm. $^{12}CH$ was completely resolved with collector slits of 200 μm for $^{13}C$ and 300 μm for $^{12}C$.

Duplicate $\delta^{13}C$ analyses of the calcite standard UWC-3 yield an external (spot to spot) precision (1σ) of 0.40–0.50‰. Mn content (shown as Mn/(Ca + Mg + Fe + Mn); Supplementary Table 1) in the calcite samples is as high as 4.6–12.7 mol. %. Such small amount of Mn has less measurable matrix effects on SIMS $\delta^{13}C$ bias than $Fe^{2+}$ [60]. Contamination of measurements by organic carbon was avoided by carefully choosing spots free of micro fractures or inclusions. Thus the potential error associated with matrix effect and organic inclusions is considered to be insignificant relative to isotopic variation.

**SIMS $\delta^{18}O_{calcite}$ analyses.** The oxygen isotope composition of calcite was measured using a SHRIMP II ion microprobe at the RSES facility of ANU. Analytical methods were similar to those described by ref. [61]. Sequences of sample analyses (typically three measurements) were bracketed by 1–2 analyses of standards. Corrected $^{18}O/^{16}O$ ratios are reported in standard $\delta^{18}O$ notation, relative to Vienna Pee Dee Belemnite (VPDB). All $\delta^{18}O$ values were calibrated against the UWC-3 calcite standard ($\delta^{18}O = -17.87$‰, 1σ = 0.03‰; cf. ref. [62]). The spot-to-spot reproducibility (i.e., external precision) was typically better than ± 0.30‰ (1σ).

**Composition and $\delta^{13}C$ of natural gas.** The composition of natural gas was determined using a Hewlett Packard 6890 II gas chromatograph (GC). Stable carbon isotope values were measured on an Optima isotope ratio mass spectrometer (IR-MS) coupled to a Hewlett Packard 6890 II GC. Measurements followed the procedure described by ref. [29]. An internal $CO_2$ reference gas was introduced into the spectrometer prior to and after the peaks of interest. Measured $\delta^{13}C$ values were calibrated using the national 'charcoal black' standard, with a value of −22.43‰ relative to VPDB. All gas samples were analyzed in triplicate.

**Biomarkers and $\delta^{13}C$ in oil samples.** The biomarker compositions of oil samples were analyzed on a Hewlett Packard 6890 II GC, using a Hewlett Packard PONA capillary column with dimensions of 50 m × 0.25 mm × 25 μm. Stable carbon isotope analyses of alkanes and arenes were performed on the same instrument, following the procedure described by ref. [63].

**Estimating the methane sink capacity of the Mahu Sag.** We first take the Mahu18-Aihu2 district, highlighted in blue dashed line in Fig. 1a, to be representative of the broader Mahu Sag. At least eight cores have been drilled in the 234 km² area, providing a good overview of the $T_1b$ reservoir strata. The combined thickness of the horizons with secondary calcite cements ranges from 19.6 m to 48.3 m, with an average value of 28.7 m. Thus, the volume of rock containing secondary calcite can be estimated as: $234 \times 10^6$ m² (area of the region of

interest) × 28.7 m (average thickness of relevant horizons) = $6.7 \times 10^9$ m³ of calcite cement-bearing strata. Multiplying by a typical sedimentary rock density of $2.5 \times 10^3$ kg m$^{-3}$ converts this volume to a total mass of $16.8 \times 10^{12}$ kg.

Based on the microscopic observations, the content of calcite cement in the relevant strata ranges from 1% to 5%, with the average value of 2.5%. Thus, we can calculate the total mass of authigenic calcite to be: $16.8 \times 10^{12}$ kg × 2.5% = $420.0 \times 10^9$ kg of calcite cement.

With the above estimate of the total mass of calcite cement in the Mahu18-Aihu2 district, we can calculate the mass of methane that would have been necessary to generate this cement. Due to its diffusion superiority of volatile natural gas relative to liquid oil, it can be approximately assumed that all carbon in the cement is derived from natural gas. As the average relative content of $CH_4$ is 85% in the $T_1b$ natural gas (Supplementary Table 2), the methane consumption can be estimated as: $420.0 \times 10^9$ kg of $CaCO_3 \div 100.09$ g mol$^{-1}$ $CaCO_3 \times 16.06$ g mol$^{-1} \times 0.85$ $CH_4 = 57.3 \times 10^9$ kg (57.3 Tg) of $CH_4$. Converting this mass to volume yields: $57.3 \times 10^9$ kg $CH_4 \div 0.72$ kg m$^{-3} \approx 79.6 \times 10^{10}$ m³ of $CH_4$.

If only taking account of the oil-charging area, a fairly conservative methane consumption is 155 Tg, as the total discovered oil-charging area has reached 633 km² to the end of 2017 in the Mahu Sag (Fig. 1a). However, authigenic calcites are not restricted in the conglomerates and sandstones charged by oil, they also occur in the wells far away from the oil-charging area. The entire sag has an area of approximately 5000 km², which is roughly 20 times greater than the area of the Mahu18-Aihu2 district. Thus, we can estimate that the total mass of $CH_4$ abiotically oxidized by Fe-Mn oxides in the sag reaches 1,224 Tg.

The parameters used in the mass balance calculations are as follows: Mass$_{CH4}$ = 16.05 g mol$^{-1}$; Mass$_{CaCO3}$ = 100.09 g mol$^{-1}$; Density of $CH_4$ = 0.72 kg m$^{-3}$ (defined at 273.15 K and 101.325 kPa); Density of rock = $2.50 \times 10^3$ kg m$^{-3}$.

## Data availability
All necessary data generated or analyzed during this study are included in this published article, and other auxiliary data are available from the corresponding authors on reasonable request.

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

## Acknowledgements

This study was supported by funds from the Natural Science Foundation of China to W.-X.H. (41830425, 41230312) and X.-L.W. (41573054), the Fundamental Research Funds for the Central Universities to X.-L.W. (020614380056), and the Outstanding PhD Candidate Program of Nanjing University to X.K. Thanks are extended to Jun Jin, Bao-li Xiang, and Zhao Yang for their assistances with core, oil and gas sampling; Ian S. Williams, Peter Holden, and Peter Lanc for their assistance with in situ stable carbon and oxygen isotope analyses; Wen-lan Zhang for her assistance with major element analyses; Huan Liu for his assistance with the analyses of Fe/Mn valence in hematite; Shane Schoepfer for improving the manuscript language.

## Author contributions

W.-X.H. and X.K. designed this study; X.K. wrote the manuscript with contributions from W.-X.H. and X.-L.W.; X.K. collected the samples and performed the experiments;

W.-X.H. developed the evaluation model for CH$_4$ sink; J.C. provided the oil and gas samples and participated in the organic geochemical experiments; X.-L.W. participated in the interpretation of the data; B.F. contributed a lot in the in situ stable isotope analyses; H.-G.W. participated in the core sampling and EPMA analyses.

## Additional information

**Competing interests:** The authors declare no competing interests.

