## [Peer Review File · Nature Communications]

Reviewers' comments:

Reviewer #1 (Remarks to the Author):

Review of the manuscript "Thermochemical oxidation of methane induced by high-valence metal oxides: insights into a deep carbon sink in sedimentary basins" by Wen-Xuan HU et al. for publication in Nature Communications.

On the basis of stable isotope and composition analyses of authigenic calcite cements, the paper proposes the first natural evidence of thermochemical oxidation of methane (TOM) by high-valence metal oxides, namely Mn and to a minor extent Fe. TOM results in highly ^{13}C depleted calcite. The addition of a CO_2 component derived from the decarboxylation of organic acids in the source rock results in less ^{13}C depleted calcite. The results and conclusions are interesting and of high importance as the authors propose that TOM may act as a major CH_4 sink in the deep crustal carbon cycle. The work describes a process never observed at such low temperatures and their conclusions will certainly open the way to further investigations on such processes in basins. The paper reads good and the English seems of good quality (I am no native English speaker). I report below my main points of concerns. I have attached a file with my minor comments and typos. Although I find the results very interesting, I am not able to judge the quality/validity of the results concerning the analyses and discussion of the hydrocarbons.

Major comments

1. MnO_2 is invoked as being the main reactant leading to oxidation of CH_4 at 80-120 °C. However it is never mentioned under what form is this MnO_2 available in the rock. Is it as an oxide or is it disseminated in minerals? Is there evidence of MnO_2 within the rock or is it transported by a fluid? I think this may have important issues regarding the possibility of the reaction at 80-120 °C. How stable is stable MnO_2 in nature?

2. In the introduction, and also in the discussion, the reactions leading to the end of the methane-zone in low-grade metamorphic conditions as reported in the external part of the Central Alps in Switzerland should also be reported (Mullis et al. 1994; Tarantola et al. 2007, 2009). CH_4 -saturated fluid is present in the marls/schists until 270 °C, temperature at which methane is oxidized during the reaction of chloritization of detrital biotite. The result is a filling of alpine fissures by calcite, however with $\delta^{13}\text{C}$ values not as low as the one reported in the present study. This reaction is also a kind of natural TOM, though at much higher temperatures than in the Junggar Basin.

Mullis J., Dubessy J., Poty B. and O'Neil J. (1994) Fluid regimes during late stages of a continental collision: physical, chemical, and stable isotope measurements of fluid inclusions in fissure quartz from a geotraverse through the Central Alps, Switzerland. *Geochimica et Cosmochimica Acta* 58(10), 2239-2267.

Tarantola A., Mullis J., Vennemann T., Dubessy J. and de Capitani C. (2007) Oxidation of methane at the $\text{CH}_4 / \text{H}_2\text{O}-(\text{CO}_2)$ transition zone in the external part of the Central Alps, Switzerland: Evidence from stable isotope investigations. *Chemical Geology* 237, 329-357.

Tarantola A., Mullis J., Guillaume D., Dubessy J., de Capitani C. and Abdelmoula M. (2009) Oxidation of CH_4 to CO_2 and H_2O by chloritization of detrital biotite at 270 ± 5 °C in the external part of the Central Alps, Switzerland. *Lithos* 112, 497-510.

3. I am not convinced with the linear correlation of Fig. 4b with R^2 of 0.28. Furthermore this correlation is never used or discussed afterwards in the text. More interesting would be to separate the two generations of calcite in Fig. 4ab in order to see if there is a difference as in Fig. 3.

4. Please precise the conditions of AOM in nature as it is written (p.11) that one of the reasons to rule out AOM is the high formation temperature >90 °C. Furthermore, in this chapter a geothermal gradient of 28.4 °C/km is assumed for all calculations. I am not sure one can be that precise in the use of a geothermal gradient. Please use a range. Same comment for the isotopic value of 4.9 ‰.

for the formation water.

5. There is an interesting discussion on the effects of duration and temperature on the reaction processes. It would be very interesting if you could provide a duration estimate leading to the oxidation of methane at 90-110 °C in this specific context.

6. I would have liked a more open conclusion with the importance of the process in any basin, rather than a brief summary of the study. How is it when there is less or no MnO₂ in the starting rock? At what conditions can we expect the process to occur when there is only Fe oxides in the rocks? Would this process also work with other high-valence oxides, at what conditions?

Minor comments

In the table caption, please label everywhere "bdl" for below detection limit.
Please check and homogenize in the tables the amount of significant digits.

The first paragraph of the introduction should also present the importance of CH₄ in the atmosphere today and the evolution of its concentration in the atmosphere during the last decades.

Fig.1. Also indicate captions for (a) and (b) parts of the figure. The figure (b) is difficult to read (at least on the printed version). I cannot distinguish muddy pebblestone from sandy gravelstone and grey muddy siltstone from grey mudstone in the lithology column. Please invert the sense of reading of the depth values. Insert a column with ages of the different members.

Fig.2. The caption for (a) indicates two generations of calcite cement whereas only one is mentioned in the text in the first line of page 5. There are two figures 2a and two figures 2b, each time the photography and the geochemical values, please separate into a, b, c and d. Please draw the two figures with the geochemical values with the same scales to ease comparisons. I cannot differentiate the circles from the diamonds on the printed version of the figure, maybe enlarge the symbols. What do you mean by high MnO content? I think "higher" is best.

Fig.8. The figure is incomplete. You must also add CO₂ derived from the decarboxylation of organic acids.

Please homogenize the conditions of temperatures of the reaction throughout the text (for example, one can read from 86.7 to 119.4 °C on p.11, from 90 to 118 °C on p. 13, from 90 to 130 °C on p.14, ...).

Supplementary material: Please use numbers with the same significant digit for the evaluation of TOM as a CH₄ sink (molar mass of CH₄ and CaCO₃). Please justify the use of a density of CH₄ of 0.72 kg/m³ ?

Nancy, July 9th 2017

Alexandre Tarantola

Reviewer #2 (Remarks to the Author):

This is a well written manuscript and should be accepted for publication. It is rather unique in that it demonstrates a methane oxidation process at moderate temperatures that has not been previously considered. The results and discussion have been strengthened by liberal use of stable isotopes of carbon and oxygen.

the only suggestion I have is that at Line 22, this seems to be the start of an "Introduction."

In the past there have been publications of shallow, low-T alteration of sediments, primarily red beds by seepage and oxidizing of methane to chemically reduce iron. This was erroneously called "bleaching." Calcium carbonate of somewhat anomalous $\delta C-13$ was produced. Examples were from Davenport field, Cement field, Ashland field, all in Oklahoma. Another is the Lisbon Valley field, Utah, and a North Sea field. There are probably microbiological processes involved in these examples. If the author would like references to these examples, they can be provided. They do NOT need to be referenced in the current manuscript.

Reviewer #3 (Remarks to the Author):

Comments on Hu et al., Thermochemical oxidation of methane induced by high-valence metal oxides: insights into a deep carbon sink in sedimentary basins

Review by Prof R H Worden, Liverpool University, UK

GENERAL COMMENTS

This is an interesting study with what seem to be good data. The paper is well written. The figures need to be improved. There is desperate need for more information on the reservoir and caprock (sedimentology, mineralogy, depositional environment, provenance, early diagenesis, thermal and burial history, etc). The paper has flaws in that it has failed to justify the supply of Mn and Fe, at high valence state (i.e. low solubility forms), to the heart of thick clastic units. The paper has failed to acknowledge vast swathes of relevant literature. It has failed to assess whether their data are normal or exceptional. It has failed to assess whether the interpretation of $\delta 13C$ patterns from other basins (and especially those with redbed caprocks and reservoirs) revealed similar patterns.

SPECIFIC COMMENTS

The opening two or three paragraph of the paper and the first sentence of the abstract are very deceptive. The paper is about methane-CO₂ relationships, I accept, but the topic has little to do with atmospheric gases. The authors appear to be trying to increase the appeal of their paper in a deceptive and scientifically-inappropriate way. There is no evidence that TSR-related processes deep in sedimentary basins have a significant effect on gas seepage to surface.

The paper ignores the vast existing literature on thermochemical sulfate reduction (TSR; a specific form of the TOM that they identify in the paper). This is a huge weakness of the paper. It is profoundly unacceptable to simply ignore a vast literature of great relevance; the following is a tiny segment of relevant papers that needed to be acknowledged (Cai et al., 2014; Jenden et al., 2015; Jiang et al., 2015; Jiang et al., 2014; Liu et al., 2013, 2014; Ma et al., 2008; Machel et al., 1994; Machel et al., 1995a; Machel et al., 1995b; Manzano et al., 1997; Worden et al., 2000; Zhang et al., 2008a; Zhang et al., 2008b). Some of these papers deal with gas-metal compound interaction and must be related to the material presented in this manuscript.

What is the evidence to support the statement on line 40-41: "TOM is potentially the predominant methane oxidation process in the subsurface environment". This has not been justified by any reference to literature. Is this a guess?

It is simply untrue (lines 41-43) to state that "However, only thermochemical sulfate reduction (TSR), which oxidizes methane relatively inefficiently, has been observed to occur in natural geological systems". Unjustified on both counts: what is the justification that TSR is inefficient (where is the evidence?) and it is not the only form of TOM. Recent papers have suggested that

REE may be involved in petroleum fluid-CO₂ relationships (Jiang et al., 2015).

What age is the Baikouquan Formation (line 57)?

A thermal and burial history of the basin and the Baikouquan Formation (line 57) must be provided to help the reader understand the thermal stress applied to the relevant rocks.

The background geology is inadequate (lines 6-67). We need to know much more about the depositional mineralogy of the reservoir unit and surrounding rocks. What was the provenance of the detrital minerals? We need to know about the early diagenesis of the reservoir and surrounding rocks. All this is needed to set the scene for the inferred later TOM-related diagenesis. It is not acceptable to state that transition metals have led to massive loss of methane without stating where they have come from (source), what happened to them during deposition and early diagenesis through to initial burial diagenesis, and why they were still in a high valence state at the time methane entered the reservoir.

The techniques used and results look to be interesting and seem to be valid. The data seem to be good and interesting.

The paper suffers from inadequate reference to the vast number of other studies on the d¹³C study of carbonate cements in clastic reservoir systems. There are hundreds of published studies; the following is a tiny sliver of the literature that the authors must access and refer to (Boles and Ramseyer, 1987; Fayek et al., 2001; Greenwood et al., 1994; Hendry et al., 1996; Irwin et al., 1977; Macaulay et al., 1998; Macaulay et al., 2000; Macaulay et al., 1993; Morad et al., 1998; Prosser et al., 1994; Ziegler et al., 2001).. Many have reported low d¹³C values. Why have these studies not been examined? It is not impossible that the authors' work could be supported by the literature; maybe others have found similar results but interpreted them differently. It is possible that the authors' work (here) may be refuted or supported but it is not acceptable to simply ignore previous work.

Are there other reasons why there may be a correlation between CH₄-d¹³C and CO₂-index shown in Fig 6? If the authors infer that only methane is involved in TOM, then why not plot CO₂/(CO₂+CH₄) instead of CO₂/(CO₂+CH_n)?

Oil data are invisible on Fig 7. Improve this figure.

I like the way the authors laid out three possible scenarios and then discounted two of them. Good, logical approach. The elevated Mn and Fe in the calcite cement seems to provide some supporting evidence for the third scenario. However, almost all calcite cements that formed during burial in all clastic reservoirs also have elevated Fe and Mn. Ferroan calcite and dolomite cements are entirely normal in deeply buried sandstones, whether they are hydrocarbon bearing or not. There is another problem: most clastic beds undergo reduction of Fe and Mn minerals during relatively early diagenesis; why do the authors regard the redbeds as being a source of high oxidation metals in the deep subsurface? Note that redbeds are typically only red at the surface: in core they are grey or white due to the normally reduced state of the transition metals in sedimentary basins. Before the authors can claim a major role for high oxidation state metals from some redbeds somewhere in the basin, they need to understand the depositional mineralogy of the reservoir unit and surrounding redbed rocks. We need to know the provenance of the detrital minerals and the depositional environment? We need to know about the early diagenesis of the reservoir and surrounding rocks. Basically, there is another study needed to reveal what reactions led to the supply of high oxidation state metals from rocks at about 100 deg C - what do the supplying rocks look like. This need does not necessarily invalidate the paper presented here but it does mean new caveats and caution need to be added.

Fig 8 is very deceptive since it ignores how the metals that do the oxidising arrive into the heart of

the reservoir unit. What is the driving force for their movement? Do they transport in high or low valence state? If the former, how is this possible given their very low aqueous solubility?

However, the reader also needs to understand whether the system that they have studied is exceptional or normal. Are the authors proposing that the scenario that they have proposed is commonplace? If this is true, then they have to refer to other analogous systems. Triassic redbed sandstones in NW Europe (and eastern N America) have redbed Upper Triassic caprocks (Ahmed, 2007; Armitage et al., 2013; Armitage et al., 2016; Burley, 1984; Gallois and Porter, 2006; Greenwood and Habesch, 1997; Hubert et al., 1992; Merino et al., 1997; Schmid et al., 2004, 2006; Strong and Milodowski, 1987; Thompson and Meadows, 1997; Wolela and Gierlowski-Kordesch, 2007): are the carbonate cements in these rocks also Mn and Fe rich? If so, then are the hydrocarbon gases characterised by altered methane $\delta^{13}C$ values?

References

- Ahmed, W., 2007, Comparison of authigenic minerals in sandstones and interbedded, mudstones, siltstones and shales, East Berlin formation, Hartford basin, USA: *Bulletin of the Chemical Society of Ethiopia*, v. 21, no. 1, p. 39-61.
- Armitage, P. J., Worden, R. H., Faulkner, D. R., Aplin, A. C., Butcher, A. R., and Espie, A. A., 2013, Mercia Mudstone Formation caprock to carbon capture and storage sites: petrology and petrophysical characteristics: *Journal of the Geological Society*, v. 170, no. 1, p. 119-132.
- Armitage, P. J., Worden, R. H., Faulkner, D. R., Butcher, A. R., and Espie, A. A., 2016, Permeability of the Mercia Mudstone: suitability as caprock to carbon capture and storage sites: *Geofluids*, v. 16, no. 1, p. 26-42.
- Boles, J. R., and Ramseyer, K., 1987, Diagenetic carbonate in Miocene sandstone reservoir, San Joaquin basin, California: *American Association of Petroleum Geologists Bulletin*, v. 71, no. 12, p. 1475-1487.
- Burley, S. D., 1984, Patterns of diagenesis in the Sherwood Sandstone Group (Triassic), United Kingdom: *Clay Minerals*, v. 19, no. 3, p. 403-440.
- Cai, C. F., He, W. X., Jiang, L., Li, K. K., Xiang, L., and Jia, L. Q., 2014, Petrological and geochemical constraints on porosity difference between Lower Triassic sour- and sweet-gas carbonate reservoirs in the Sichuan Basin: *Marine and Petroleum Geology*, v. 56, p. 34-50.
- Fayek, M., Harrison, T. M., Grove, M., McKeegan, K. D., Coath, C. D., and Boles, J. R., 2001, In situ stable isotopic evidence for protracted and complex carbonate cementation in a petroleum reservoir, North Coles Levee, San Joaquin Basin, California, USA: *Journal of Sedimentary Research*, v. 71, no. 3, p. 444-458.
- Gallois, R. W., and Porter, R. J., 2006, The stratigraphy and sedimentology of the Dunscombe Mudstone Formation (Late Triassic) of Southwest England, in Pirrie, D., ed., *Geoscience in South-West England*, Vol 11, Pt 3, 2006, Volume 11, p. 174-182.
- Greenwood, P. J., and Habesch, S. M., 1997, Diagenesis of the Sherwood Sandstone Group in the southern East Irish Sea Basin (Blocks 110/13, 110/14 and 110/15): constraints from preliminary isotopic and fluid inclusion studies, in Meadows, N. S., Trueblood, S. P., Hardman, M., and Cowan, G., eds., *Petroleum Geology of the Irish Sea and Adjacent Areas*, Volume 124, p. 353-371.
- Greenwood, P. J., Shaw, H. F., and Fallick, A. E., 1994, Petrographic and isotopic evidence for diagenetic processes in Middle Jurassic sandstones and mudrocks from the Brae area, North Sea: *Clay Minerals*, v. 29, no. 4, p. 637-650.
- Hendry, J. P., Trewin, N. H., and Fallick, A. E., 1996, Low-Mg calcite marine cement in Cretaceous turbidites: Origin, spatial distribution and relationship to seawater chemistry: *Sedimentology*, v. 43, no. 5, p. 877-900.
- Hubert, J. F., Feshbachmeriney, P. E., and Smith, M. A., 1992, The Triassic-Jurassic Hartford Rift Basin, Connecticut and Massachusetts - evolution, sandstone diagenesis and hydrocarbon history: *American Association of Petroleum Geologists Bulletin*, v. 76, no. 11, p. 1710-1734.
- Irwin, H., Curtis, C., and Coleman, M. L., 1977, Isotopic evidence for source of diagenetic carbonates formed during burial of organic rich sediments: *Nature*, v. 269, p. 209-213.

- Jenden, P. D., Titley, P. A., and Worden, R. H., 2015, Enrichment of nitrogen and ^{13}C of methane in natural gases from the Khuff Formation, Saudi Arabia, caused by thermochemical sulfate reduction: *Organic Geochemistry*, v. 82, p. 54-68.
- Jiang, L., Cai, C. F., Worden, R. H., Li, K. K., Xiang, L., Chu, X. L., Shen, A. J., and Li, W. J., 2015, Rare earth element and yttrium (REY) geochemistry in carbonate reservoirs during deep burial diagenesis: Implications for REY mobility during thermochemical sulfate reduction: *Chemical Geology*, v. 415, p. 87-101.
- Jiang, L., Worden, R. H., and Cai, C. F., 2014, Thermochemical sulfate reduction and fluid evolution of the Lower Triassic Feixianguan Formation sour gas reservoirs, northeast Sichuan Basin, China: *American Association of Petroleum Geologists Bulletin*, v. 98, no. 5, p. 947-973.
- Liu, Q. Y., Worden, R. H., Jin, Z. J., Liu, W. H., Li, J., Gao, B., Zhang, D. W., Hu, A. P., and Yang, C., 2013, TSR versus non-TSR processes and their impact on gas geochemistry and carbon stable isotopes in Carboniferous, Permian and Lower Triassic marine carbonate gas reservoirs in the Eastern Sichuan Basin, China: *Geochimica et Cosmochimica Acta*, v. 100, p. 96-115.
- , 2014, Thermochemical sulphate reduction (TSR) versus maturation and their effects on hydrogen stable isotopes of very dry alkane gases: *Geochimica et Cosmochimica Acta*, v. 137, p. 208-220.
- Ma, Q. S., Ellis, G. S., Amrani, A., Zhang, T. W., and Tang, Y. C., 2008, Theoretical study on the reactivity of sulfate species with hydrocarbons: *Geochimica et Cosmochimica Acta*, v. 72, no. 18, p. 4565-4576.
- Macaulay, C. I., Fallick, A., McLaughlin, O. M., Haszeldine, R. S., and Pearson, M. J., 1998, The significance of $\delta^{13}\text{C}$ of carbonate cement in reservoir sandstones: a regional perspective from the Jurassic of the Northern North Sea: In: *Carbonate cementation in sandstones* (ed. Morad, S.) International Association of Sedimentologists Special Publications, v. 26, p. 395-408.
- Macaulay, C. I., Fallick, A. E., Haszeldine, R. S., and McAulay, G. E., 2000, Oil migration makes the difference: regional distribution of carbonate cement $\delta^{13}\text{C}$ in northern North Sea Tertiary sandstones: *Clay Minerals*, v. 35, no. 1, p. 69-76.
- Macaulay, C. I., Haszeldine, R. S., and Fallick, A. E., 1993, Distribution, chemistry, isotopic composition and origin of diagenetic carbonates - Magnus Sandstone, North Sea: *Journal of Sedimentary Petrology*, v. 63, no. 1, p. 33-43.
- Machel, H. G., Krouse, H. R., Foght, J. M., Riciputi, L. R., and Cole, D. R., 1994, The Devonian Nisku sour gas play, Alberta, Canada - a unique natural laboratory for the study of thermochemical sulfate reduction: *Abstracts of Papers of the American Chemical Society*, v. 208, p. 106-GEOC.
- Machel, H. G., Krouse, H. R., Riciputi, L. R., and Cole, D. R., 1995a, Devonian Nisku sour gas play, Canada: A unique natural laboratory for study of thermochemical sulfate reduction, in Vairavamurthy, M. A., and Schoonen, M. A. A., eds., *Geochemical Transformations of Sedimentary Sulfur*, Volume 612, p. 439-454.
- Machel, H. G., Krouse, H. R., and Sassen, R., 1995b, Products and distinguishing criteria of bacterial and thermochemical sulfate reduction: *Applied Geochemistry*, v. 10, no. 4, p. 373-389.
- Manzano, B. K., Fowler, M. G., and Machel, H. G., 1997, The influence of thermochemical sulphate reduction on hydrocarbon composition in Nisku reservoirs, Brazeau river area, Alberta, Canada: *Organic Geochemistry*, v. 27, no. 7-8, p. 507-521.
- Merino, E., Girard, J. P., May, M. T., and Ranganathan, V., 1997, Diagenetic mineralogy, geochemistry, and dynamics of mesozoic arkoses, Hartford rift basin, Connecticut, USA: *Journal of Sedimentary Research*, v. 67, no. 1, p. 212-224.
- Morad, S., De Ros, L. F., Nystuen, J. P., and Bergan, M., 1998, Carbonate diagenesis and porosity evolution in sheet-flood sandstones: evidence from the Middle and Lower Members (Triassic) in the Snorre Field, Norwegian North Sea, in Morad, S., ed., *Carbonate cementation in sandstones*. International Association of Sedimentologists Special Publication, Volume 26: Oxford, Blackwells, p. 53-85.
- Prosser, D. J., Fallick, A. E., Daws, J. A., and Williams, B. P. J., 1994, Geochemistry and diagenesis of stratabound calcite cement layers within the Rannoch Formation of the Brent Group, Murchison Field, North Viking Graben (Northern North Sea) - Reply: *Sedimentary Geology*, v. 93, no. 1-2, p. 143-147.
- Schmid, S., Worden, R. H., and Fisher, Q. J., 2004, Diagenesis and reservoir quality of the

- Sherwood Sandstone (Triassic), Corrib Field, Slyne Basin, west of Ireland: *Marine and Petroleum Geology*, v. 21, no. 3, p. 299-315.
- , 2006, Carbon isotope stratigraphy using carbonate cements in the Triassic Sherwood Sandstone Group: Corrib Field, west of Ireland: *Chemical Geology*, v. 225, no. 1-2, p. 137-155.
- Strong, G. E., and Milodowski, A. E., 1987, Aspects of the diagenesis of the Sherwood Sandstones of the Wessex Basin and their influence on reservoir characteristics, in Marshall, J. D., ed., *Diagenesis of sedimentary sequences*, Volume 36: London, Geological Society of London, p. 325-337.
- Thompson, J., and Meadows, N. S., 1997, Clastic sabkhas and diachroneity at the top of the Sherwood Sandstone Group: East Irish Sea Basin, in Meadows, N. S., Trueblood, S. P., Hardman, M., and Cowan, G., eds., *Petroleum Geology of the Irish Sea and Adjacent Areas*, Volume 124, p. 237-251.
- Wolela, A. M., and Gierlowski-Kordesch, E. H., 2007, Diagenetic history of fluvial and lacustrine sandstones of the Hartford Basin (Triassic-Jurassic), Newark Supergroup, USA: *Sedimentary Geology*, v. 197, no. 1-2, p. 99-126.
- Worden, R. H., Smalley, P. C., and Cross, M. M., 2000, The influence of rock fabric and mineralogy on thermochemical sulfate reduction: Khuff Formation, Abu Dhabi: *Journal of Sedimentary Research*, v. 70, no. 5, p. 1210-1221.
- Zhang, T. W., Amrani, A., Ellis, G. S., Ma, Q. S., and Tang, Y. C., 2008a, Experimental investigation on thermochemical sulfate reduction by H₂S initiation: *Geochimica et Cosmochimica Acta*, v. 72, no. 14, p. 3518-3530.
- Zhang, T. W., Ellis, G. S., Walters, C. C., Kelemen, S. R., Wang, K. S., and Tang, Y. C., 2008b, Geochemical signatures of thermochemical sulfate reduction in controlled hydrous pyrolysis experiments: *Organic Geochemistry*, v. 39, no. 3, p. 308-328.
- Ziegler, K., Coleman, M. L., and Howarth, R. J., 2001, Palaeohydrodynamics of fluids in the Brent Group (Oseberg Field, Norwegian North Sea) from chemical and isotopic compositions of formation waters: *Applied Geochemistry*, v. 16, no. 6, p. 609-632.

Detailed Response to Reviewers

Itemized and detailed responses ('R') to the reviewers' comments are appended below. Key revisions have been highlighted by blue in the revised manuscript for easy tracking.

1. Response to Reviewer 1# (Alexandre Tarantola)

Many thanks for your careful review and constructive comments on our manuscript.

(1) MnO_2 is invoked as being the main reactant leading to oxidation of CH_4 at 80-120 °C. However it is never mentioned under what form is this MnO_2 available in the rock. Is it as an oxide or is it disseminated in minerals? Is there evidence of MnO_2 within the rock or is it transported by a fluid? I think this may have important issues regarding the possibility of the reaction at 80-120 °C. How stable is stable MnO_2 in nature?

R: We carried out more experiments to investigate the occurrence of high-valence Mn/Fe minerals. Results have been added to the revised manuscript (Lines 2–11, Page 14) and Supplementary Materials (from Line 11 on Page 6 to Line 14 on Page 8).

Abundant brown mudstones and muddy conglomerates occur in the Baikouquan Formation (Fig. S3). These rocks can be the source of high-oxidation-state metals in the deep subsurface. To determine the types and occurrences of such high-valence Mn/Fe minerals, we performed XRD, FE-SEM, SEM-EDS, and EPMA analyses on the brown mudstones and brown pebbly conglomerate. Results indicate that high-valence Mn does not occur as separate MnO_2 (e.g., pyrolusite, ramsdellite). Mn exists in hematite, which is widespread in the matrix of brown conglomerates and sandstones, as well as in the brown mudstones. As shown in Fig. R1 (or S4), hematite occurs as isolated amorphous

aggregates in the matrix, with an average Mn_2O_3 content of 1.14 wt.%, or is disseminated in the clay
 with an average content of 0.68 wt.%. Considering the lower permeability of brown mudstone, we
 propose that the high-oxidation-state metals came mainly from the matrix of reservoir rocks during
 TOM, rather than being transported from surrounding mudstones or underlying strata by fluids.

*In situ* XPS (X-ray photoelectron spectroscopy) spectra of high-valence Mn/Fe minerals were
 collected to investigate the valence of Mn. Unfortunately, effective Mn 2p and 3s signals were not
 obtained due to the low Mn content (Galakhov et al., 2002; Ilton et al., 2016). However, obvious Fe
 2p spectra indicate that Fe is trivalent in the hematite (Fig. R1d or S4d). In the presence of Fe^{3+} , Mn
 generally occurs as an isomorphous $Mn^{3+/4+}$ substitution in hematite (Alvarez et al., 2007; Liu et al.,
 2016; Manceau et al., 2000; Singh et al., 2002).

a

Spot No.	1	2	3	4	5	6
	hematite				hematite+smectite	
CaO	0.01	bdl	bdl	0.02	0.64	0.50
Na ₂ O	0.04	0.02	bdl	0.09	0.09	0.30
K ₂ O	0.04	0.01	0.02	0.06	1.23	3.56
TiO ₂	1.16	2.05	1.99	3.03	3.68	1.03
MgO	0.10	0.03	bdl	0.01	0.53	2.95
Fe ₂ O ₃	95.49	95.16	94.25	91.44	60.26	41.64
Al ₂ O ₃	0.28	0.39	0.38	0.56	11.18	13.58
Mn ₂ O ₃ *	1.15	0.95	1.00	1.47	0.61	0.74
SiO ₂	1.21	1.30	1.27	1.94	20.44	28.30
Total	99.45	99.89	98.91	98.61	98.66	92.59

*bdl"denotes below detection limit.

c

b

d

11
 12 **Fig. R1.** Minerals containing high-valence manganese in the T₁b reservoir rocks. (a) Isolated hematite containing a

1 high Mn content and hematite disseminated in the clay-rich matrix. (b) XRD spectrum showing abundant hematite
in the smectite and I/S rich matrix of the conglomerate. (c) Mn content shown as Mn_2O_3 of the isolated hematite is
obviously higher than that of the hematite disseminated in the clay. The data are obtained from EPMA. (d) XPS
spectra indicate the iron is trivalent in the hematite of the brown rock; the Fe^{2+} signal is from ilmenite.

In weakly acidic fluids, $\text{Mn}^{3+/4+}$ is released into the formation water through the dissolution of
hematite (Artamonova et al., 2013; Walanda et al., 2005). The subsequent reduction of $\text{Mn}^{3+/4+}$ also
promotes its dissolution and release. However, aqueous Mn^{3+} is unstable and tends to undergo
disproportionation to Mn^{2+} and MnO_2 (Walanda et al., 2005; Armstrong, 2008). In contrast with
insoluble MnO_2 ($E_h = 1.224 \text{ V}$), aqueous Mn^{3+} ($E_h = 1.542 \text{ V}$) is characterized by a stronger
oxidization capacity (Haynes, 2014). This may explain why methane can be oxidized at such low
temperatures.

(2) In the introduction, and also in the discussion, the reactions leading to the end of the
methane-zone in low-grade metamorphic conditions as reported in the external part of the Central
Alps in Switzerland should also be reported (Mullis et al. 1994; Tarantola et al. 2007, 2009).
CH_4 -saturated fluid is present in the marls/schists until $270 \text{ }^\circ\text{C}$, temperature at which methane is
oxidized during the reaction of chloritization of detrital biotite. The result is a filling of alpine
fissures by calcite, however with $\delta^{13}\text{C}$ values not as low as the one reported in the present study.
This reaction is also a kind of natural TOM, though at much higher temperatures than in the Junggar
Basin.

**R:** Agreed. The oxidation of methane during the chloritization of detrital biotite is also a form
of TOM occurring at $>270^\circ\text{C}$. The papers by Mullis et al. (1994) and Tarantola et al. (2007, 2009) are

cited in the revised manuscript (line 5–6, page 3).

(3) I am not convinced with the linear correlation of Fig. 4b with R2 of 0.28. Furthermore this
correlation is never used or discussed afterwards in the text. More interesting would be to separate
the two generations of calcite in Fig. 4a, b in order to see if there is a difference as in Fig. 3.

**R:** The linear correlation (Fig. 4b) has been removed from the revised manuscript.

As the two stages of calcites are inter-fingered, it is difficult to separate them. The $\delta^{13}\text{C}$ and
$\delta^{18}\text{O}$ values (Figs 4a, b) were obtained from whole-rock analyses. As a result, these two plots
describe the mixed $\delta^{13}\text{C}$ and $\delta^{18}\text{O}$ signatures of two-stage calcites in each sample. In the revised
manuscript, we mention *in situ* carbon and oxygen isotopic analyses by secondary-ion mass
spectroscopy (SIMS). As shown in Fig. 4c, late-stage calcite cements are generally characterized by
lower $\delta^{13}\text{C}$ values than those of the early-stage calcite. Please see the revised Fig. 4 for detailed
information.

(4) Please precise the conditions of AOM in nature as it is written (p.11) that one of the reasons
to rule out AOM is the high formation temperature $>90\text{ }^\circ\text{C}$. Furthermore, in this chapter a geothermal
gradient of $28.4\text{ }^\circ\text{C}/\text{km}$ is assumed for all calculations. I am not sure one can be that precise in the
use of a geothermal gradient. Please use a range. Same comment for the isotopic value of $-4.9\text{ }‰$ for
the formation water.

**R:** Agreed. The precise requirements for AOM in nature include (a) a sufficient supply of
methane; (b) the survival of archaeal groups such as ANME-1, ANME-2, and ANME-3 (Kallmeyer

& Boetius, 2004; Knittel et al., 2005; Orphan et al., 2001); (c) the existence of an electron acceptor
such as MnO_2 , SO_4^{2-} , or Fe^{3+} (Beal et al., 2009; Sivan et al., 2014; Wenger et al., 2002); (d) low
formation temperatures of $<80^\circ\text{C}$ (Kallmeyer & Boetius, 2004; Peter, 2005); and (e) other favorable
conditions, such as the presence of N, P, and K, relatively low salinity, and suitable acidity (Wenger
et al., 2002).

The paleogeothermal gradient does vary temporally and spatially. During the first methane
charging (J_1), the paleogeothermal gradient in the Mahu Sag decreased from 32 to 28°C km^{-1} (Qiu et
al., 2002). During this period, the formation temperature is estimated to have been 83°C – 122°C , as
mentioned in the revised manuscript (Fig. R2 or S5). Please see lines 5–8 in page 12 of the revised
manuscript.

**Fig. R2.** The burial history and thermal evolution of the Baikouquan Formation in the study area, taking Well Ma18
as an example. The basic stratigraphic sequence and tectonic–geothermal evolution of the study area are from Qi et
al. (2015) and Qiu et al. (2002), respectively.

Formation-water $\delta^{18}\text{O}$ values in clastic strata gradually increase with enhanced interactions

between silicate minerals and water (Egeberg and Aagaard, 1989; Wilkinson et al., 1992). The
diagenetic characteristics of the Baikouquan Formation are similar to those of the Brae Formation
sandstone (North Sea), where the $\delta^{18}\text{O}$ of the formation water is reported to be -6‰ to -4‰
(Marchand et al., 2002). Therefore, we used a $\delta^{18}\text{O}$ range of -6‰ to -4‰ to calculate the
precipitation temperature of calcite. Results indicate that the precipitation of Mn-rich calcite
occurred at $90\text{--}130^\circ\text{C}$ (Fig. R3). T_1b is characterized by a high temperature ($>80^\circ\text{C}$) during oil and
gas charging, and the metabolism of microorganisms is significantly inhibited at such high
temperatures.

Please see lines 9–13, page 11, for detailed information.

**Fig. R3.** Calculated precipitation temperature of authigenic calcites in the T_1b reservoir rocks.

(5) There is an interesting discussion on the effects of duration and temperature on the reaction
processes. It would be very interesting if you could provide a duration estimate leading to the
oxidation of methane at $90\text{--}110^\circ\text{C}$ in this specific context.

**R:** As shown in Fig. R2, the formation temperature was $>80^\circ\text{C}$ during methane charging in the
Lower Jurassic (*ca* 185 Ma). At such temperatures, methane might have been very slowly oxidized
by $\text{Mn}^{3+/4+}$. When the formation temperature reached 90°C (*ca* 175 Ma), methane would have been

oxidized more efficiently, and the reaction has continued until the present.

To constrain crystallization temperature of calcites, the homogeneous temperature (Th) of
primary fluid inclusions were tested. However, it is really a tough work searching fluid inclusions in
calcite cements in T_{1b}, as they are so small. Through two weeks of tentative experiments, we
eventually obtained 8 valid values from 32 double-side polished sections. The Th values are 97°C,
101 °C, 105 °C, 107 °C, 112 °C, 116 °C, 123 °C, 126 °C, within 90–130 °C overall and with an
average of 110.9 °C. While as the data points are so limited that we ultimately decided to not show
them in the revised manuscript.

An estimated duration were added in the revised manuscript (line 6–8, page 15).

**Fig. R4.** The occurrence of primary fluid inclusions and their homogeneous temperature in authigenic calcites of
T_{1b}.

(6) I would have liked a more open conclusion with the importance of the process in any basin,
rather than a brief summary of the study. How is it when there is less or no MnO₂ in the starting rock?
At what conditions can we expect the process to occur when there is only Fe oxides in the rocks?
Would this process also work with other high-valence oxides, at what conditions?

**R:** Regarding the first question, the reactions between methane and Mn^{3+/4+}, Fe³⁺, and SO₄²⁻ are
thermodynamically possible at elevated temperatures. When there is little or no MnO₂ in the clastic
rock, Fe³⁺ can be an effective electron acceptor in TOM, as reported for the oxidation of methane

during the chloritization of detrital biotite (Mullis et al., 1994; Tarantola et al., 2007, 2009). However,
the reaction between methane and Fe^{3+} tends to occur at higher temperatures due to its high
activation energy (Beal et al., 2009).

Regarding the second question, the occurrence of abundant Fe-rich carbonate cements in deeply
buried hydrocarbon-bearing sandstones indicates that Fe^{3+} was involved in the oxidation of
hydrocarbons at 100–150°C (Boles and Ramseyer, 1987; Fayek et al., 2001; Greenwood et al., 1994;
Macaulay et al., 1993, 1998, 2000; Morad et al., 1998; Prosser et al., 1994). This temperature is
lower than that in the studies for the oxidation of methane during chloritization of detrital biotite
(~270°C; Mullis et al., 1994; Tarantola et al., 2007, 2009). Recently, we are conducting relevant
thermal simulation experiments in silica capillary tubes, combined with Laser Raman Spectrum
detection. Results indicate that CO_2 signal was detected by 1,100 min, during the oxidation of CH_4
by MnO_2 (pyrolusite) at 200°C involved in water. At similar conditions, CO_2 was detected by 1,566
13 min using Fe_2O_3 (hematite) at 250°C. Moreover, similar to the TSR process, the supply of aqueous
Fe^{3+} also constrains the start of the reaction and the reduction efficiency of methane (Worden et al.,
2000). We propose that the temperature and solubility of Fe^{3+} species provide significant constraints
on TOM involving Fe^{3+} .

As to the last question, other high-valence oxidants can also cause the oxidation of methane.
The most common is SO_4^{2-} , with methane being oxidized at temperatures of >135 °C when the
$\text{C}_1/\text{C}_{1-4}$ ratio is >90% (Jenden et al., 2015; Jiang et al., 2014, 2015; Krouse et al., 1988; Liu et al.,
2013, 2014; Worden & Smalley, 1996). In addition, it is proposed that some REE may also cause the
oxidation of methane (Jiang et al., 2016), but more work is needed to confirm this possibility.

(7) The seven minor comments mainly on the table and figures.

**R:** We revised the tables, figures, and related expressions according to the seven comments.

The detailed information is as follows.

(a) In the data tables, all results below the detection limit are labeled as “bdl”, and the number
of significant figures is now consistent throughout.

(b) The concentration of methane (CH₄) has increased from only 680 ppbv in pre-industrial
8 times to 1,779 ppbv in 2010 (Kirschke et al., 2013; Hopcroft et al., 2017). Coupled with its relatively
high radiative efficiency, methane is the second most important anthropogenic greenhouse gas in the
atmosphere (Boucher et al., 2009). Please see Line 6–8 in Page 2 in the revised manuscript.

(c) Fig. 1 was improved to distinguish different rocks in the lithologic column, and a column
showing the age of T_{1b} was added (we did not find specific ages of different members in previous
studies).

(d) In the revised Fig. 2, the photographs and geochemical values are separated into Fig. 2a–d,
and geochemical values are shown on the same scales for ease comparison. The figure is also
improved to distinguish circles from diamonds. In the caption, the text is modified to present both
generations of calcite cements (Lines 4–8 on Page 6).

(e) Regarding Fig. 8, CO₂ derived from the decarboxylation of organic acids has been added,
and the oxidant "MnO₂" changed to "Mn^{3+/4+}, (Fe³⁺) in hematite".

(f) Considering both the formation temperature and calculated calcite precipitation temperature,
the temperatures of the reaction throughout the text are made consistent at 90°C–130°C.

(g) In the supplementary material for the evaluation of CH₄ amount consumed by TOM, the significant digit of molar mass of CH₄ and CaCO₃ was changed to the same number. The CH₄ density value of 0.72 kg/m³ is defined at 0 °C (273.15 K) and 1 atm (101.325 kPa). Relevant information was added to the Supplemental Materials in the Line 4-5 of Page 22 of the revised manuscript.

Reference

1. Alvarez, M., Rueda, E.H., Sileo, E.E. (2007). Simultaneous incorporation of Mn and Al in the goethite structure. *Geochimica et Cosmochimica Acta*, 71(4), 1009–1020.
2. Armstrong, F. A. (2008). Why did Nature choose manganese to make oxygen?. *Philosophical Transactions of the Royal Society of London B: Biological Sciences*, 363(1494), 1263–1270.
3. Artamonova, I. V., Gorichev, I. G., Godunov, E. B. (2013). Kinetics of manganese oxides dissolution in sulphuric acid solutions containing oxalic acid. *Engineering*, 5(09), 714–719.
4. Beal, E. J., House, C. H., Orphan, V. J. (2009). Manganese- and iron-dependent marine methane oxidation. *Science*, 325(5937), 184–187.
5. Boles, J.R., Ramseyer, K. (1987). Diagenetic carbonate in Miocene sandstone reservoir, San Joaquin basin, California. *AAPG Bulletin*, 71(12), 1475–1487.
6. Boucher, O., Friedlingstein, P., Collins, B., Shine, K. P. (2009). The indirect global warming potential and global temperature change potential due to methane oxidation. *Environmental Research Letters*, 4(4), 044007.
7. Egeberg, P. K., Aagaard, P. (1989). Origin and evolution of formation waters from oil fields on the Norwegian shelf. *Applied Geochemistry*, 4(2), 131–142.
8. Fayek, M., Harrison, T.M., Grove, M., McKeegan, K.D., Coath, C.D., Boles, J.R. (2001). In situ stable isotopic evidence for protracted and complex carbonate cementation in a petroleum reservoir, North Coles Levee, San Joaquin Basin, California, USA. *Journal of Sedimentary Research*, 71(3), 444–458.

-
- 9. Galakhov, V. R., Demeter, M., Bartkowski, S., Neumann, M., Ovechkina, N. A., Kurmaev, E. Z.,
Lobachevskaya, N. I., Mukovskii, Ya. M., Mitchell, J., Ederer, D. L. (2002). Mn 3s exchange splitting in
mixed-valence manganites. *Physical Review B*, 65(113102), 1–4.
- 10. Greenwood, P. J., Shaw, H. F., Fallick, A. E. (1994). Petrographic and isotopic evidence for diagenetic
processes in Middle Jurassic sandstones and mudrocks from the Brae area, North Sea. *Clay Minerals*, 29(4),
637–650.
- 11. Haynes, W. M. (Ed.). (2014). *CRC handbook of chemistry and physics*. CRC press, 958–967.
- 12. Hopcroft, P. O., Valdes, P. J., O'Connor, F. M., Kaplan, J. O., Beerling, D. J. (2017). Understanding the glacial
methane cycle. *Nature Communications*, 8(14383), 1–10.
- 13. Ilton, E. S., Post, J. E., Heaney, P. J., Ling, F. T., Kerisit, S. N. (2016). XPS determination of Mn oxidation
states in Mn (hydr) oxides. *Applied Surface Science*, 366, 475–485.
- 14. Jenden, P. D., Titley, P. A., Worden, R. H. (2015). Enrichment of nitrogen and ¹³C of methane in natural gases
from the Khuff Formation, Saudi Arabia, caused by thermochemical sulfate reduction. *Organic Geochemistry*,
82, 54–68.
- 15. Jiang, L., Worden, R. H., Cai, C. F. (2014). Thermochemical sulfate reduction and fluid evolution of the Lower
Triassic Feixianguan Formation sour gas reservoirs, northeast Sichuan Basin, China. *AAPG Bulletin*, 98(5),
947–973.
- 16. Jiang, L., Cai, C.F., Worden, R.H., Li, K.K., Xiang, L., Chu, X.L., Shen, A.J., Li, W.J. (2015). Rare earth
element and yttrium (REY) geochemistry in carbonate reservoirs during deep burial diagenesis: Implications
for REY mobility during thermochemical sulfate reduction. *Chemical Geology*, 415, 87–101.
- 17. Kallmeyer, J., Boetius, A. (2004). Effects of temperature and pressure on sulfate reduction and anaerobic
oxidation of methane in hydrothermal sediments of Guaymas Basin. *Applied and environmental microbiology*,
70(2), 1231–1233.
- 18. Kirschke, S., Bousquet, P., Ciais, P., Saunoy, M., Canadell, J. G., Dlugokencky, E. J., ..., Cameron-Smith, P.
(2013). Three decades of global methane sources and sinks. *Nature Geoscience*, 6(10), 813–823.
- 19. Knittel, K., Lösekann, T., Boetius, A., Kort, R., Amann, R. (2005). Diversity and distribution of
methanotrophic archaea at cold seeps. *Applied and environmental microbiology*, 71(1), 467–479.
- 20. Krouse, H.R., Viau, C.A., Eliuk, L.S., Ueda, A., Halas, S. (1988). Chemical and isotopic evidence of
thermochemical sulphate reduction by light hydrocarbon gases in deep carbonate reservoirs. *Nature*,
333(6172), 415–419.

-
- 21. Liu, H., Lu, X., Li, J., Chen, X., Zhu, X., Xiang, W., Zhang, R., Wang, X., Lu, J., Wang, R. (2017).
Geochemical fates and unusual distribution of arsenic in natural ferromanganese duricrust. *Applied*
*Geochemistry*, 76, 74–87.
- 22. Liu, Q.Y., Worden, R.H., Jin, Z.J., Liu, W.H., Li, J., Gao, B., Zhang, D.W., Hu, A.P., Yang, C. (2013). TSR
versus non-TSR processes and their impact on gas geochemistry and carbon stable isotopes in Carboniferous,
Permian and Lower Triassic marine carbonate gas reservoirs in the Eastern Sichuan Basin, China. *Geochimica*
*et Cosmochimica Acta*, 100, 96–115.
- 23. Liu, Q. Y., Worden, R. H., Jin, Z. J., Liu, W. H., Li, J., Gao, B., Zhang, D.W., Hu, A. P., Yang, C. (2014).
Thermochemical sulphate reduction (TSR) versus maturation and their effects on hydrogen stable isotopes of
very dry alkane gases. *Geochimica et Cosmochimica Acta*, 137, 208–220.
- 24. Macaulay, C. I., Haszeldine, R. S., Fallick, A. E. (1993). Distribution, chemistry, isotopic composition and
origin of diagenetic carbonates—Magnus Sandstone, North Sea. *Journal of Sedimentary Petrology*, 63(1),
33–43.
- 25. Macaulay, C.I., Fallick, A., McLaughlin, O.M., Haszeldine, R.S., Pearson, M.J. (1998). The significance of
$\delta^{13}\text{C}$ of carbonate cement in reservoir sandstones: a regional perspective from the Jurassic of the Northern
North Sea, in Morad, S., ed., *Carbonate cementation in sandstones*. International Association of
Sedimentologists Special Publications, 26, 395–408.
- 26. Macaulay, C.I., Fallick, A.E., Haszeldine, R.S., McAulay, G.E. (2000). Oil migration makes the difference:
regional distribution of carbonate cement $\delta^{13}\text{C}$ in northern North Sea Tertiary sandstones. *Clay Minerals*, 35(1),
69–76.
- 27. Manceau, A., Schlegel, M.L., Musso, M., Sole, V.A., Gauthier, C., Petit, P.E., Trolard, F. (2000). Crystal
chemistry of trace elements in natural and synthetic goethite. *Geochimica et Cosmochimica Acta*, 64(21),
3643–3661.
- 28. Marchand, A. M., Macaulay, C. I., Haszeldine, R. S., Fallick, A. E. (2002). Pore water evolution in oilfield
sandstones: constraints from oxygen isotope microanalyses of quartz cement. *Chemical Geology*, 191(4),
285–304.
- 29. Morad, S., De Ros, L. F., Nystuen, J. P., Bergan, M. (1998). Carbonate diagenesis and porosity evolution in
sheet-flood sandstones: evidence from the Middle and Lower Members (Triassic) in the Snorre Field,
Norwegian North Sea, in Morad, S., ed., *Carbonate cementation in sandstones*. International Association of
Sedimentologists Special Publication, 26, 53–85.

-
- 30. Mullis, J., Dubessy, J., Poty, B., O'Neil, J. (1994). Fluid regimes during late stages of a continental collision:
physical, chemical, and stable isotope measurements of fluid inclusions in fissure quartz from a geotraverse
through the Central Alps, Switzerland. *Geochimica et Cosmochimica Acta*, 58(10), 2239–2267.
- 31. Orphan, V. J., House, C. H., Hinrichs, K. U., McKeegan, K. D., DeLong, E. F. (2001). Methane-consuming
archaea revealed by directly coupled isotopic and phylogenetic analysis. *Science*, 293(5529), 484–487.
- 32. Peters, K. E., Walters, C. C., Moldowan, J. M. (2005). *The biomarker guide*, Cambridge University Press,
645–705.
- 33. Prosser, D.J., Fallick, A.E., Daws, J.A., Williams, B.P.J. (1994). Geochemistry and diagenesis of stratabound
calcite cement layers within the Rannoch Formation of the Brent Group, Murchison Field, North Viking
Graben (Northern North Sea)—Reply. *Sedimentary Geology*, 93(1–2), 143–147.
- 34. Qiu, N.S., Yang, H.B., Wang, X.L. (2002). Tectono-thermal evolution in the Junggar Basin. *Chinese Journal of*
*Geology*, 37, 423–429.
- 35. Qi, W., Pan, J., Wang, G., Qu, Y., Tan, K., Yin, L. (2015). Fluid inclusion and hydrocarbon charge history for
reservoir of Baikouquan Formation in the Mahu Sag, Junggar Basin. *Natural Gas Geoscience*, 26, 64–71 (In
Chinese with English abstract).
- 36. Singh, B., Sherman, D.M., Gilkes, R.J., Wells, M.A., Mosselmans, J.F.W. (2002). Incorporation of Cr, Mn and
Ni into goethite (α-FeOOH): mechanism from extended X-ray absorption fine structure spectroscopy. *Clay*
*Minerals*, 37(4), 639–649.
- 37. Sivan, O., Antler, G., Turchyn, A. V., Marlow, J. J., Orphan, V. J. (2014). Iron oxides stimulate sulfate-driven
anaerobic methane oxidation in seeps. *Proceedings of the National Academy of Sciences*, 111(40), 4139–4147.
- 38. Tarantola A., Mullis J., Vennemann T., Dubessy J., de Capitani C. (2007). Oxidation of methane at the
CH₄/H₂O-(CO₂) transition zone in the external part of the Central Alps, Switzerland: Evidence from stable
isotope investigations. *Chemical Geology*, 237(3), 329–357.
- 39. Tarantola A., Mullis J., Guillaume D., Dubessy J., de Capitani C., Abdelmoula M. (2009). Oxidation of CH₄ to
CO₂ and H₂O by chloritization of detrital biotite at 270 ± 5 °C in the external part of the Central Alps,
Switzerland. *Lithos*, 112(3), 497–510.
- 40. Walanda, D. K., Lawrance, G. A., Donne, S. W. (2005). Hydrothermal MnO₂: synthesis, structure,
morphology and discharge performance. *Journal of Power Sources*, 139(1), 325–341.

-
- 41. Wenger, L. M., Isaksen, G. H. (2002). Control of hydrocarbon seepage intensity on level of biodegradation in
sea bottom sediments. *Organic Geochemistry*, 33(12), 1277–1292.
- 42. Wilkinson, M., Crowley, S. F., Marshall, J. D. (1992). Model for the evolution of oxygen isotope ratios in the
pore fluids of mudrocks during burial. *Marine and Petroleum Geology*, 9(1), 98–105.
- 43. Worden, R. H., Smalley, P. C. (1996). H₂S-producing reactions in deep carbonate gas reservoirs: Khuff
Formation, Abu Dhabi. *Chemical Geology*, 133(1–4), 157–171.
- 44. Worden, R.H., Smalley, P.C., Oxtoby, N.H. (1996). The effects of thermochemical sulfate reduction upon
formation water salinity and oxygen isotopes in carbonate gas reservoirs. *Geochimica et Cosmochimica Acta*,
60(20), 3925–3931.

2. Response to Reviewer 2# (Anonymous Reviewer)

(1) The suggestion I have is that at Line 22, this seems to be the start of an "Introduction."

R: We also wanted to change the manuscript in this way, but remaining the "Introduction" is against journal requirement.

(2) In the past there have been publications of shallow, low-T alteration of sediments, primarily red beds by seepage and oxidizing of methane to chemically reduce iron. This was erroneously called "bleaching." Calcium carbonate of somewhat anomalous $\delta^{13}\text{C}$ was produced. Examples were from Davenport field, Cement field, Ashland field, all in Oklahoma. Another is the Lisbon Valley field, Utah, and a North Sea field. There are probably microbiological processes involved in these examples. If the author would like references to these examples, they can be provided. They do NOT need to be referenced in the current manuscript.

R: We investigated a few papers on "bleaching" sandstones (Beitler et al., 2003; Chan et al., 2000, 2004; Parry et al., 2004, 2009; Wigley et al., 2012). Considering that the $\delta^{13}\text{C}$ values of carbonate cements in these sandstones were not extremely low, and that the values are probably responses to microbiological processes, these papers are not cited in the revised manuscript.

Reference

1. Beitler, B., Chan, M. A., Parry, W. T. (2003). Bleaching of Jurassic Navajo sandstone on Colorado Plateau Laramide highs: Evidence of exhumed hydrocarbon supergiants?. *Geology*, 31(12), 1041–1044.
2. Chan, M. A., Parry, W. T., Bowman, J. R. (2000). Diagenetic hematite and manganese oxides and fault-related fluid flow in Jurassic sandstones, southeastern Utah. *AAPG Bulletin*, 84(9), 1281–1310.

-
- 3. Chan, M. A., Beitler, B., Parry, W. T., Ormö, J., Komatsu, G. (2004). A possible terrestrial analogue for
haematite concretions on Mars. *Nature*, 429(6993), 731–734.
- 4. Parry, W. T., Chan, M. A., Beitler, B. (2004). Chemical bleaching indicates episodes of fluid flow in
deformation bands in sandstone. *AAPG Bulletin*, 88(2), 175–191.
- 5. Parry, W. T., Chan, M. A., Nash, B. P. (2009). Diagenetic characteristics of the Jurassic Navajo Sandstone in
the Covenant oil field, central Utah thrust belt. *AAPG Bulletin*, 93(8), 1039–1061.
- 6. Wigley, M., Kampman, N., Dubacq, B., Bickle, M. (2012). Fluid-mineral reactions and trace metal
mobilization in an exhumed natural CO₂ reservoir, Green River, Utah. *Geology*, 40(6), 555–558.

3. Response to Reviewer 3# (Prof. Richard H. Worden)

Thanks for your careful review and detailed comments on our manuscript. The literature you recommended is important but we cannot cite all the papers, as the maximum allowed number of references in the journal is 70.

(1) The opening two or three paragraph of the paper and the first sentence of the abstract are very deceptive. The paper is about methane-CO₂ relationships, I accept, but the topic has little to do with atmospheric gases. The authors appear to be trying to increase the appeal of their paper in a deceptive and scientifically-inappropriate way. There is no evidence that TSR-related processes deep in sedimentary basins have a significant effect on gas seepage to surface.

R: In the revised manuscript, the title has been changed to “Thermochemical oxidation of methane induced by high-valence metal oxides in a deep sedimentary basin”. We also removed most related discussion of implications of the greenhouse effect. Please see the revised abstract and introduction.

Previous studies have reported numerous methane seeps in sedimentary basins and suggested that these are important sources of greenhouse gases (Berbesi et al., 2014; Etiope and Klusman, 2002; Etiope et al., 2008; Klusman et al., 1998; Kroeger, et al., 2011; Zheng et al., 2017, 2018). Moreover, in the Fourth Assessment Report of the Intergovernmental Panel on Climate Change (IPCC), geological methane was listed as an important greenhouse gas (Etiope, 2015). In the Junggar Basin, more than 200 oil and gas seeps have been discovered since the beginning of oil/gas exploration at the end of the 19th century (Zheng et al., 2018). The nearest oil seepage is only ~20 km from our study area (Wu et al., 2002; Zheng et al., 2018). In addition, many mud volcanoes occur at the

1 southern margin of the Junggar Basin, one of which produced a crater lake of dimensions 120 m by
2 35 m during an eruption in 1995 (Dai et al., 2012; Ma et al., 2014; Zheng et al., 2017). More than 91%
of the natural gas from this mud volcano is methane (Dai et al., 2012). During its dormant period, the
total CH₄ emission is at least 22.6 tons per year (Zheng et al., 2017). Thermochemical oxidation of
methane might therefore be an important means of consuming methane in sedimentary basins or in
the deep crust. We have retained a few sentences regarding the effect of TOM on methane sinks in
the revised manuscript (lines 5–10, page 2).

(2) The paper ignores the vast existing literature on thermochemical sulfate reduction (TSR; a
specific form of the TOM that they identify in the paper). This is a huge weakness of the paper. It is
profoundly unacceptable to simply ignore a vast literature of great relevance; the following is a tiny
segment of relevant papers that needed to be acknowledged (Cai et al., 2014; Liu et al., 2013, 2014;
12 Ma et al., 2008; Machel et al., 1994; Machel et al., 1995a, b; Manzano et al., 1997; Worden et al.,
2000; Zhang et al., 2008a; Zhang et al., 2008b). Some of these papers deal with gas-metal compound
interaction and must be related to the material presented in this manuscript.

**R:** Agreed. We have cited most of the relevant papers in the revised manuscript. Please see
paragraphs 2 and 3 in the revised Introduction (from lines 22 page 3 to line 2 page 3).

We had studied some of these papers already, but focused on the extremely ¹³C-depleted
carbonate cements. The papers were helpful in improving our understanding of the occurrence of
TOM in the Baikouquan Formation, particularly concerning, for example, the oxidation of methane
in association with the occurrence and supply of electron-acceptor materials, as mentioned in your
comments below.

(3) What is the evidence to support the statement on line 40-41: "TOM is potentially the
predominant methane oxidation process in the subsurface environment". This has not been justified
by any reference to literature. Is this a guess?

**R:** Previous studies have shown that AOM-related microorganisms survive at temperatures of
$<80^{\circ}\text{C}$. In the revised manuscript, this sentence has been changed to "TOM is potentially the
predominant methane oxidation process in the deep subsurface with a high formation temperature
($>80^{\circ}\text{C}$)" (lines 20–22, page 2). To further support this statement, we added more citations (Jenden et
al., 2015; Krouse et al., 1988; Machel, 2001; Peter, 2005; Wenger et al., 2002; Worden & Smalley,
1996; Worden et al., 2000).

(4) It is simply untrue (lines 41-43) to state that "However, only thermochemical sulfate
reduction (TSR), which oxidizes methane relatively inefficiently, has been observed to occur in
natural geological systems". Unjustified on both counts: what is the justification that TSR is
inefficient (where is the evidence?) and it is not the only form of TOM. Recent papers have
suggested that REE may be involved in petroleum fluid- CO_2 relationships (Jiang et al., 2015).

**R:** We intended to state that methane is oxidized relatively inefficiently in the presence of
abundant C_{2+} components because these would be preferentially oxidized (Hao et al., 2008; Liu et al.,
2013, 2014). Of course, methane can act as an effective electron-donor during TSR in dry gas
reservoirs (Cai et al., 2004; Krouse et al., 1988; Worden et al., 1996; Worden & Smalley, 1996; Liu et
al., 2013; Jenden et al., 2015). To avoid further confusion, we removed this sentence from the revised
manuscript.

(5) What age is the Baikouquan Formation (line 57)?

**R:** The Baikouquan Formation was deposited during the Lower Triassic (Lei et al., 2015; Jia et
al, 2016) at *ca* 252–247 Ma (International Commission on Stratigraphy, 2017).

Please see line 14 of page 3 and the revised Fig. 1b for detailed information.

(6) A thermal and burial history of the basin and the Baikouquan Formation (line 57) must be
provided to help the reader understand the thermal stress applied to the relevant rocks.

**R:** Agreed. The burial history and thermal evolution of the Baikouquan Formation are now
presented in the section “Geological Setting” in the Supplementary Materials. Please see Fig. S5 in
the Supplementary Materials. Also, please refer to our response to question 4 of Reviewer #1.

(7) The background geology is inadequate (lines 6-67). We need to know much more about the
depositional mineralogy of the reservoir unit and surrounding rocks. What was the provenance of the
detrital minerals? We need to know about the early diagenesis of the reservoir and surrounding rocks.
All this is needed to set the scene for the inferred later TOM-related diagenesis. It is not acceptable to
state that transition metals have led to massive loss of methane without stating where they have come
from (source), what happened to them during deposition and early diagenesis through to initial burial
diagenesis, and why they were still in a high valence state at the time methane entered the reservoir.

**R:** All of the information is now presented in the the Supplementary Geological Settings. The
mineralogy of the reservoir unit and surrounding rocks is presented in subsection 5, “Mineralogy and
Occurrence of Mn(fe) oxides” (Lines 12–20 on Page 6); the provenance of detrital minerals is
discussed in subsection 4, “Provenance” (from Line 9 on Page 5 to Line 10 on Page 6); the early
diagenesis of the reservoir and surrounding rocks of T_{1b} is summarized in subsection 7, “Burial

history and Diagenesis” (from Line 15 on Page 9 to Line 9 on Page 10); and the occurrence of
high-valence Mn(Fe) oxides (source) is described in subsection 5, “Mineralogy and Occurrence of
Mn(fe) oxides” (from Line 21 on Page 6 to Line 14 on Page 8).

As to “what happened to them during deposition and early diagenesis through to initial burial
diagenesis, and why they were still in a high valence state at the time methane entered the reservoir?”,
the occurrence of abundant subsurface brown mudstones and conglomerates in T_{1b} (Fig. S3)
indicates the depletion of organic matter during deposition, and the weak decomposition of organic
matter during early diagenesis. Therefore, the high-valence Mn/Fe oxides were not reduced before
methane entered the reservoir. Please see relevant descriptions in subsection 3, “Sedimentary
petrology and Depositional Environment” (from Line 11 on Page 3 to Line 8 on Page 5), and
subsection 7, “Burial history and Diagenesis” (Line 6–9 on Page 10).

(8) The paper suffers from inadequate reference to the vast number of other studies on the $\delta^{13}\text{C}$
study of carbonate cements in clastic reservoir systems. Many have reported low $\delta^{13}\text{C}$ values. Why
have these studies not been examined? It is not impossible that the authors' work could be supported
by the literature; maybe others have found similar results but interpreted them differently. It is
possible that the authors' work (here) may be refuted or supported but it is not acceptable to simply
ignore previous work.

**R:** We found that ^{13}C -depleted carbonate cements have been reported in many clastic reservoirs,
such as the Middle Jurassic reservoir sandstones in the northern North Sea (to -28.8% VPDB,
Macaulay et al., 1998) and the Miocene Surma Group sandstones in the Surma Basin of Bangladesh
(-23.1% , Rahman & McCann, 2012). However, almost all of the relevant studies suggested that the

low $\delta^{13}\text{C}$ values were caused by the thermal decarboxylation of organic matter (Boles and Ramseyer,
1987; Fayek et al., 2001; Irwin et al., 1977; Liu et al., 2017; Macaulay et al., 1993, 1998, 2000;
Morad et al., 1998; Prosser et al., 1994; Rahman & McCann, 2012). Regarding the extremely low
$\delta^{13}\text{C}$ values (less than -25‰), such as those reported for the late calcite cement in the Sherwood
Sandstone Group in the southern East Irish Sea Basin (Greenwood and Habesch, 1997), a value of
6 -30.4‰ might be partly ascribed to the thermochemical oxidation of hydrocarbon gases. In the
7 revised manuscript, we have cited more previous studies and added a relevant discussion. Please see
lines 8–13 of page 9 in the revised manuscript.

(9) Are there other reasons why there may be a correlation between $\text{CH}_4\text{-}\delta^{13}\text{C}$ and $\text{CO}_2\text{-index}$
shown in Fig 6? If the authors infer that only methane is involved in TOM, then why not plot
$\text{CO}_2/(\text{CO}_2+\text{CH}_4)$ instead of $\text{CO}_2/(\text{CO}_2+\text{CH}_n)$?

**R:** Thermal cracking of hydrocarbons might lead to a similar correlation between $\text{CH}_4\text{-}\delta^{13}\text{C}$ and
the CO_2 index in Fig. 6. During thermal oil cracking with water involvement, with $\text{Ro} < 2.0\%$, the
yields of CH_4 and CO_2 increase with increasing temperature (Ma et al., 2016; Galimov, 2006;
Seewald, 2003; Tian et al., 2012). $\text{CH}_4\text{-}\delta^{13}\text{C}$ and $\text{C}_2\text{H}_6\text{-}\delta^{13}\text{C}$ values generally increase at the same
time (Ma et al., 2016; Galimov, 2006; Tian et al., 2012). As a result, both $\text{CH}_4\text{-}\delta^{13}\text{C}$ and $\text{C}_2\text{H}_6\text{-}\delta^{13}\text{C}$
values become heavier with increasing $\text{CO}_2\text{-index}$. However, this correlation between $\delta^{13}\text{C}\text{-C}_{2-4}$ and
$\text{CO}_2/(\text{CO}_2+\text{CH}_n)$ was not observed in our study area. Therefore, the obvious positive correlation
between $\text{CH}_4\text{-}\delta^{13}\text{C}$ and $\text{CO}_2\text{-index}$ indicates that methane is the major reactant during the reduction
of high-valence Mn/Fe oxides in T_1b .

Fig. 6 shows the correlations between $\text{CH}_4\text{-}\delta^{13}\text{C}$ and $\text{CO}_2\text{-index}$, and between $\text{C}_{2-4}\text{-}\delta^{13}\text{C}$ and

CO₂-index, revealing that CH₄-δ¹³C increases with increasing CO₂-index, while the C₂₋₄-δ¹³C vs
CO₂-index plot does not show such a trend. This result supports our conclusion that methane was the
dominant reactant in the reduction of high-valence Mn/Fe oxides in T_{1b}. For this reason, the
correlation between δ¹³C-C₂₋₄ and CO₂/(CO₂+CH_n) is also plotted in Fig. 6.

(10) Oil data are invisible on Fig 7. Improve this figure.

**R:** Agreed. We have revised the figure accordingly.

(11) Calcite cements that formed during burial in all clastic reservoirs also have elevated Fe and
Mn. Ferroan calcite and dolomite cements are entirely normal in deeply buried sandstones, whether
they are hydrocarbon bearing or not. There is another problem: most clastic beds undergo reduction
of Fe and Mn minerals during relatively early diagenesis; why do the authors regard the redbeds as
being a source of high oxidation metals in the deep subsurface? Note that redbeds are typically only
red at the surface: in core they are grey or white due to the normally reduced state of the transition
metals in sedimentary basins. Before the authors can claim a major role for high oxidation state
metals from some redbeds somewhere in the basin, they need to understand the depositional
mineralogy of the reservoir unit and surrounding redbed rocks. We need to know the provenance of
the detrital minerals and the depositional environment? We need to know about the early diagenesis
of the reservoir and surrounding rocks. Basically, there is another study needed to reveal what
reactions led to the supply of high oxidation state metals from rocks at about 100 deg C - what do the
supplying rocks look like. This need does not necessarily invalidate the paper presented here but it
does mean new caveats and caution need to be added.

**R:** It is true that Fe- and Mn-rich calcite and dolomite cements commonly occur in deeply

buried sandstones. As Mn^{2+} and Fe^{2+} are soluble in the formation water, they can be transported
during fluid convection and migration (Garven et al., 1985; Bjørlykke, 1993, 1994, 2015; Kharaka &
Hanor, 2003; Miall, 2013), and their source in Fe- and Mn-rich carbonates is not solely related to
hydrocarbon-related fluids.

However, abundant brown mudstones and conglomerates can be observed in T_1b core samples
(Fig. S3), and these types of rocks might be the source of high-oxidation-state metals in the deep
subsurface. Further mineralogical analyses indicate that Mn does not occur as separate MnO_2 (e.g.,
pyrolusite, ramsdellite), and that high-valence manganese exists in hematite, which is widespread in
the matrix of brown conglomerates and sandstones, as well as in the brown mudstones. As shown in
Fig. S4, hematite occurs as isolated amorphous aggregates in the matrix, with an average Mn_2O_3
content of 1.14 wt.%, or is disseminated in the clay with an average Mn_2O_3 content of 0.68 wt.%.
Considering the lower permeability of brown mudstone, we propose that the high-oxidation-state
metals were derived mainly from the matrix of reservoir rocks during TOM.

The T_1b was derived from two areas with different lithological components: the basin basement
comprising granite and mafic–ultramafic igneous rocks, and underlying Carboniferous–Permian
sedimentary sequences (Zheng et al., 2000; Cao et al., 2005). Please see the section on “Provenance”
(from Line 9 on Page 5 to Line 10 on Page 6) in the Supplementary Geological Settings.

T_1b was deposited in a succession of clastic delta fan systems in an oxic-lacustrine environment
(Jia et al., 2016). Details are given in the subsection “Sedimentary petrology and depositional
environment” (from Line 11 on Page 3 to Line 8 on Page 5) of the Supplementary Geological

Settings.

The early diagenesis of the reservoir and surrounding rocks of T_1b is summarized in the
subsection “Early diagenesis” (from Line 15 on Page 9 to Line 9 on Page 10) of the Supplementary
Geological Settings.

With regard to “what reactions led to the supply of high oxidation state metals from rocks at
about 100°C and what do the supplying rocks look like?”, the dissolution of hematite in weakly
acidic formation water (Artamonova et al., 2013; Kang et al., 2017; Walanda et al., 2005) and the
promotion of reduction by methane could supply high-oxidation-state Mn(Fe) for the slow thermal
oxidation of methane. As mentioned above, the original brown reservoir rocks are the main source of
high-oxidation-state Mn(Fe) (Fig. S3).

(12) Fig 8 is very deceptive since it ignores how the metals that do the oxidizing arrive into the
heart of the reservoir unit. What is the driving force for their movement? Do they transport in high or
low valence state? If the former, how is this possible given their very low aqueous solubility?

**R:** The occurrence of high-valence Mn oxides is closely associated with the start of oxidation of
methane at 90–130°C. The high-valence Mn(Fe) oxides are distributed in the matrix among grains of
reservoir rocks, such as brown muddy pebbly conglomerates. These oxides are within the reservoir
rocks, instead of being transported in fluids from surrounding mudstones. When methane-bearing
fluids from underlying Permian source rocks charged the T_1b reservoir sequence, methane
encountered high-valence Mn(Fe) and reduced it to $Mn^{2+}(Fe^{2+})$ under favorable conditions.
$Mn^{2+}(Fe^{2+})$ are soluble and would be transported in formation water, resulting in the precipitation of

Mn-rich calcites in the presence of CO_3^{2-} .

Please also see our responses to question 1 raised by reviewer 1.

(13) However, the reader also needs to understand whether the system that they have studied is
exceptional or normal. Are the authors proposing that the scenario that they have proposed is
commonplace? If this is true, then they have to refer to other analogous systems. Triassic redbed
sandstones in NW Europe (and eastern N America) have redbed Upper Triassic caprocks. Are the
carbonate cements in these rocks also Mn and Fe rich? If so, then are the hydrocarbon gases
characterised by altered methane $\delta^{13}\text{C}$ values?

**R:** The precipitation of Mn-rich calcite cements with extremely low $\delta^{13}\text{C}$ values in the T_{1b}
sandy conglomerate reservoir is closely related to the following relatively unique geological system.

(a) Abundant mafic–ultramafic rocks occur in the provenance area, and weathering in an arid–
semiarid paleoclimate is important for the supply of abundant high-oxidation-state metals.

(b) Organic matter became depleted in the Baikouquan Formation during deposition;
consequently, high-valence Mn oxides were preserved during early diagenesis.

(c) Detrital carbonate, playa-lacustrine carbonate, and calcrete that formed during the
synsedimentary stage are not observed in the Baikouquan Formation, and if such “inorganic”
carbonate minerals exist, the effect of hydrocarbon/water/rock interactions on the $\delta^{13}\text{C}$ value of
carbonate cement would be strongly buffered.

(d) Thermochemical oxidation of methane is favored at high formation temperatures ($>80^\circ\text{C}$).

The paleoclimate during deposition of the Lower Triassic Sherwood Sandstone Group (SSG)

was similar to that of the T₁b reservoir. Abundant redbeds and playa sediments in northwest Europe
were deposited in relatively high-salinity water, which resulted in the precipitation of calcrite and
dolocrete in near-surface sediments (Greenwood and Habesch, 1997; Schmid et al., 2006a, b; Strong
and Milodowski, 1987; Thompson and Meadows, 1997). Despite buffering by early-diagenetic
carbonates with high $\delta^{13}\text{C}$ ($\sim -3\text{‰}$ to 2‰ ; Schmid et al., 2006b), the late calcite still has very low
$\delta^{13}\text{C}$ values between -16.6‰ and -30.4‰ (VPDB). Values of less than -25‰ may be derived from
a thermally altered hydrocarbon gas source (Cai et al., 2014; Irwin et al., 1977; Krouse et al., 1988;
Worden et al., 1996a, b, 2000; Jiang et al., 2014, 2015). In the Triassic–Jurassic Newark Supergroup
of the Hartford Basin, the strata contain playa sediments and calcrite (Ahmed, 2007; Hubert et al.,
1992; Gierlowski-Kordesch, 1998; Wolela and Gierlowski-Kordesch, 2007). These
deposition-related carbonate minerals would influence the $\delta^{13}\text{C}$ signatures of carbonate cements
associated with the oxidation of hydrocarbons.

The thermochemical oxidation of methane might have occurred in some CH₄-bearing redbed
sandstones, but the extremely low $\delta^{13}\text{C}$ values of carbonate cements are not easy to measure. We
speculate that much lower $\delta^{13}\text{C}$ values could be detected in late carbonate cements via *in situ* carbon
isotope analysis, by SIMS for example.

With regard to “Are the carbonate cements in these rocks also Mn and Fe rich? If so, then are
the hydrocarbon gases characterised by altered methane $\delta^{13}\text{C}$ values?”, Mn-bearing ferroan calcite
and dolomite were observed in the Sherwood Sandstone Group (Greenwood and Habesch, 1997;
Strong and Milodowski, 1987), but we did not find published data on hydrocarbon gases in the strata.
If methane was significantly oxidized in these clastic rocks, CH₄- $\delta^{13}\text{C}$ and CH₄- δD values would

increase with increasing $\text{CO}_2/(\text{CO}_2+\text{C}_n\text{H}_{2n+2})$, and the range of $(\delta^{13}\text{C}_{\text{C}_2\text{H}_6} - \delta^{13}\text{C}_{\text{CH}_4})$ values would be
reduced, possibly even overturning the generally increasing $\text{C}_1\text{--C}_4$ $\delta^{13}\text{C}$ trend (Krouse et al., 1988;
Worden et al., 1996; Hao et al., 2008; Liu et al., 2013, 2014). The $\text{C}_1\text{--C}_4$ δD trend might be also
overturned, together with the consumption of methane (Liu et al., 2014). Of course, the types of
hydrocarbon-generating organic matter and thermal maturity must be taken into account during such
investigations.

Reference

- 1. Ahmed, W. (2007). Comparison of authigenic minerals in sandstones and interbedded, mudstones, siltstones
and shales, East Berlin formation, Hartford basin, USA. *Bulletin of the Chemical Society of Ethiopia*, 21(1),
39–61.
- 2. Artamonova, I.V., Gorichev, I.G., Godunov, E.B. (2013). Kinetics of manganese oxides dissolution in
sulphuric acid solutions containing oxalic acid. *Engineering*, 5(09), 714–719.
- 3. Berbesi, L. A., di Primio, R., Anka, Z., Horsfield, B., Wilkes, H. (2014). Methane leakage from evolving
petroleum systems: Masses, rates and inferences for climate feedback. *Earth and Planetary Science Letters*
387, 219–228.
- 4. Bjørlykke, K. (1994). Fluid-flow processes and diagenesis in sedimentary basins, in Parnell, J., ed., *Geofluids:*
*Origin, Migration and Evolution of Fluids in Sedimentary Basins. Geological Society London Special*
*Publication*, 78, 127–140.
- 5. Bjørlykke, K. (2015). Heat transport in sedimentary basins. *Petroleum Geoscience*, Springer, 273–277.
- 6. Boles, J.R., Ramseyer, K. (1987). Diagenetic carbonate in Miocene sandstone reservoir, San Joaquin basin,
California. *AAPG Bulletin*, 71(12), 1475–1487.
- 7. Cai, C., Xie, Z., Worden, R. H., Hu, G., Wang, L., He, H. (2004). Methane-dominated thermochemical
sulphate reduction in the Triassic Feixianguan Formation East Sichuan Basin, China: towards prediction of
fatal H_2S concentrations. *Marine and Petroleum Geology*, 21(10), 1265–1279.
- 8. Cai, C. F., He, W. X., Jiang, L., Li, K. K., Xiang, L., Jia, L. Q. (2014). Petrological and geochemical
constraints on porosity difference between Lower Triassic sour- and sweet-gas carbonate reservoirs in the
Sichuan Basin. *Marine and Petroleum Geology*, 56, 34–50.
- 9. Cao, J., Zhang, Y.J., Hu, W.X., Yao, S.P., Wang, X.L., Zhang, Y.Q., Tang, Yong. (2005). The Permian hybrid
petroleum system in the northwest margin of the Junggar Basin, northwest China. *Marine and Petroleum*
*Geology*, 22(3), 331–349.
- 10. Dai, J., Wu, X., Ni, Y., Wang, Z., Zhao, C., Wang, Z., Liu, G. (2012). Geochemical characteristics of natural
gas from mud volcanoes in the southern Junggar Basin. *Science China Earth Sciences*, 55(3), 355–367 (In
Chinese with English abstract).

-
- 11. Etiope, G., Klusman, R. W. (2002). Geologic emissions of methane to the atmosphere. *Chemosphere* 49,
777–789.
- 12. Etiope, G., Milkov, A. V., Derbyshire, E. (2008). Did geologic emissions of methane play any role in
Quaternary climate change?. *Global and Planetary Change*, 61(1), 79–88.
- 13. Etiope G., 2015. *Natural Gas Seepage: The Earth's Hydrocarbon Degassing*. Springer, 1–15.
- 14. Fayek, M., Harrison, T.M., Grove, M., McKeegan, K.D., Coath, C.D., Boles, J.R. (2001). In situ stable
isotopic evidence for protracted and complex carbonate cementation in a petroleum reservoir, North Coles
Levee, San Joaquin Basin, California, USA. *Journal of Sedimentary Research*, 71(3), 444–458.
- 15. Galimov, E. M. (2006). Isotope organic geochemistry. *Organic Geochemistry*, 37(10), 1200–1262.
- 16. Garven, G. (1985). The role of regional fluid flow in the genesis of the Pine Point deposit, western Canada
sedimentary basin. *Economic Geology*, 80(2), 307–324.
- 17. Gierlowski-Kordesch, E.H. (1998). Carbonate deposition in an ephemeral siliciclastic alluvial system: Jurassic
Shuttle Meadow Formation, Newark Supergroup, Hartford basin, USA. *Palaeogeography, Palaeoclimatology,*
*Palaeoecology*, 140(1), 161–184.
- 18. Greenwood, P.J., Habesch, S.M., Meadows, N.S., Trueblood, S.P., Hardman, M., Cowan, G. (1997).
Diagenesis of the Sherwood Sandstone Group in the southern East Irish Sea Basin (Blocks 110/13, 110/14 and
110/15): constraints from preliminary isotopic and fluid inclusion studies. *Petroleum Geology of the Irish Sea*
*and Adjacent Areas*, 124, 353–371.
- 19. Hao, F., Guo, T., Zhu, Y., Cai, X., Zou, H., Li, P. (2008). Evidence for multiple stages of oil cracking and
thermochemical sulfate reduction in the Puguang gas field, Sichuan Basin, China. *AAPG Bulletin*, 92(5),
611–637.
- 20. Hubert, J.F., Feshbachmeriney, P.E., Smith, M.A. (1992). The Triassic-Jurassic Hartford Rift Basin,
Connecticut and Massachusetts—evolution, sandstone diagenesis and hydrocarbon history. *AAPG Bulletin*,
76(11), 1710–1734.
- 21. Irwin, H., Curtis, C., Coleman, M.L. (1977). Isotopic evidence for source of diagenetic carbonates formed
during burial of organic rich sediments. *Nature*, 269(5625), 209–213.
- 22. Jenden, P. D., Titley, P. A., and Worden, R. H. (2015). Enrichment of nitrogen and ¹³C of methane in natural
gases from the Khuff Formation, Saudi Arabia, caused by thermochemical sulfate reduction. *Organic*
*Geochemistry*, 82, 54–68.
- 23. Jia, H., Ji, H., Li, X., Zhou, H., Wang, L., Gao, Y. (2016). A retreating fan-delta system in the northwestern
Junggar Basin, northwestern China—Characteristics, evolution and controlling factors. *Journal of Asian Earth*
*Sciences*, 123, 162–177.
- 24. Jia, H., Ji, H., Li, X., Zhou, H., Wang, L., Gao, Y. A. (2016). Retreating fan-delta system in the northwestern
Junggar Basin, northwestern China—Characteristics, evolution and controlling factors. *Journal of Asian Earth*
*Sciences*, 123, 162–177.
- 25. Jiang, L., Worden, R. H., and Cai, C. F. (2014). Thermochemical sulfate reduction and fluid evolution of the
Lower Triassic Feixianguan Formation sour gas reservoirs, northeast Sichuan Basin, China. *AAPG Bulletin*,
98(5), 947–973.
- 26. Jiang, L., Cai, C.F., Worden, R.H., Li, K.K., Xiang, L., Chu, X.L., Shen, A.J., Li, W.J. (2015). Rare earth
element and yttrium (REY) geochemistry in carbonate reservoirs during deep burial diagenesis: Implications
for REY mobility during thermochemical sulfate reduction. *Chemical Geology*, 415, 87–101.

-
- 27. Kang, X., Hu, W.X., Cao, J., Jin, J., Wu, H., Zhao, Y., Wang, J. (2017). Selective dissolution of alkali feldspars
and its effect on Lower Triassic sandy conglomerate reservoirs in the Junggar Basin, northwestern China.
*Geological Journal*, DOI: 10.1002/gj.2905.
- 28. Kelley, D. S., Früh-Green, G. L. (1999). Abiogenic methane in deep-seated mid-ocean ridge environments:
Insights from stable isotope analyses. *Journal of Geophysical Research: Solid Earth*, 104(B5), 10439–10460.
- 29. Kharaka, Y. K., Hanor, J. S. (2003). Deep fluids in the continents: I. Sedimentary basins. *Treatise on*
*Geochemistry*, 5, 499–540.
- 30. Klusman, R. W., Jakel, M. E. (1998). Natural microseepage of methane to the atmosphere from the
Denver-Julesburg basin, Colorado. *Journal of Geophysical Research: Atmospheres*, 103(D21), 28041–28045.
- 31. Kroeger, K. F., di Primio, R., Horsfield, B. (2011). Atmospheric methane from organic carbon mobilization in
sedimentary basins—The sleeping giant?. *Earth-Science Reviews*. 107(3), 423–442.
- 32. Lei, D., Abulimiti, Tang, Y., Chen, J., Cao, J. (2014). Controlling Factors and Occurrence Prediction of High
Oil-Gas Production Zones in Lower Triassic Baikouquan Formation of Mahu Sag in Junggar Basin. *Xinjiang*
*Petroleum Geology*, 35, 495–499 (In Chinese with English abstract).
- 33. Liu, Q.Y., Worden, R.H., Jin, Z.J., Liu, W.H., Li, J., Gao, B., Zhang, D.W., Hu, A.P., Yang, C. (2013). TSR
versus non-TSR processes and their impact on gas geochemistry and carbon stable isotopes in Carboniferous,
Permian and Lower Triassic marine carbonate gas reservoirs in the Eastern Sichuan Basin, China. *Geochimica*
*et Cosmochimica Acta*, 100, 96–115.
- 34. Liu, Q. Y., Worden, R. H., Jin, Z. J., Liu, W. H., Li, J., Gao, B., Zhang, D.W., Hu, A. P., Yang, C. (2014).
Thermochemical sulphate reduction (TSR) versus maturation and their effects on hydrogen stable isotopes of
very dry alkane gases. *Geochimica et Cosmochimica Acta*, 137, 208–220.
- 35. Ma, A. (2016). Kinetics of oil-cracking for different types of marine oils from Tahe Oilfield, Tarim Basin, NW
China. *Journal of Natural Gas Geoscience*, 1(1), 35–43.
- 36. Ma, X., Zheng, G., Guo, Z., Etiopie, G., Fortin, D., Sano, Y. (2014). Estimation of greenhouse gas flux from
mud volcanoes in the Dushanzi area, southern Junggar Basin of Northwest China. *Chinese Science Bulletin*,
59(32), 3190–3196.
- 37. Macaulay, C.I., Fallick, A., McLaughlin, O.M., Haszeldine, R.S., Pearson, M.J. (1998). The significance of
$\delta^{13}\text{C}$ of carbonate cement in reservoir sandstones: a regional perspective from the Jurassic of the Northern
North Sea, in Morad, S., ed., *Carbonate cementation in sandstones*. International Association of
Sedimentologists Special Publications, 26, 395–408.
- 38. Macaulay, C. I., Haszeldine, R. S., and Fallick, A. E. (1993). Distribution, chemistry, isotopic composition and
origin of diagenetic carbonates—Magnus Sandstone, North Sea. *Journal of Sedimentary Petrology*, 63(1),
33–43.
- 39. Macaulay, C.I., Fallick, A.E., Haszeldine, R.S., McAulay, G.E. (2000). Oil migration makes the difference:
regional distribution of carbonate cement $\delta^{13}\text{C}$ in northern North Sea Tertiary sandstones. *Clay Minerals*, 35(1),
69–76.
- 40. Machel, H. G. (2001). Bacterial and thermochemical sulfate reduction in diagenetic settings—old and new
insights. *Sedimentary Geology*, 140(1), 143–175.
- 41. Morad, S., De Ros, L. F., Nystuen, J. P., Bergan, M. (1998). Carbonate diagenesis and porosity evolution in
sheet-flood sandstones: evidence from the Middle and Lower Members (Triassic) in the Snorre Field,
Norwegian North Sea, in Morad, S., ed., *Carbonate cementation in sandstones*. International Association of
Sedimentologists Special Publication, 26, 53–85.
- 42. Mullis, J., Dubessy, J., Poty, B., O'Neil, J. (1994). Fluid regimes during late stages of a continental collision:
physical, chemical, and stable isotope measurements of fluid inclusions in fissure quartz from a geotraverse
through the Central Alps, Switzerland. *Geochimica et Cosmochimica Acta*, 58(10), 2239–2267.

-
- 43. Peters, K. E., Walters, C. C., Moldowan, J. M. (2005). *The biomarker guide*, Cambridge University Press,
645–705.
- 44. Prosser, D.J., Fallick, A.E., Daws, J.A., Williams, B.P.J. (1994). Geochemistry and diagenesis of stratabound
calcite cement layers within the Rannoch Formation of the Brent Group, Murchison Field, North Viking
Graben (Northern North Sea)—Reply. *Sedimentary Geology*, 93(1–2), 143–147.
- 45. Rahman, M.J.J., McCann, T. (2012). Diagenetic history of the Surma group sandstones (Miocene) in the
Surma Basin, Bangladesh. *Journal of Asian Earth Sciences*, 45, 65–78.
- 46. Seewald, J. S. (2003). Organic–inorganic interactions in petroleum-producing sedimentary basins. *Nature*,
426(6964), 327–333.
- 47. Schmid, S., Worden, R.H., Fisher, Q.J. (2004). Diagenesis and reservoir quality of the Sherwood Sandstone
(Triassic), Corrib Field, Slyne Basin, west of Ireland. *Marine and Petroleum Geology*, 21(3), 299–315.
- 48. Schmid, S., Worden, R.H., Fisher, Q.J. (2006). Carbon isotope stratigraphy using carbonate cements in the
Triassic Sherwood Sandstone Group: Corrib Field, west of Ireland. *Chemical Geology*, 225(1–2). 137–155.
- 49. Strong, G.E., Milodowski, A.E. (1987). Aspects of the diagenesis of the Sherwood Sandstones of the Wessex
Basin and their influence on reservoir characteristics, in Marshall, J. D., ed., *Diagenesis of sedimentary*
*sequences*. Geological Society of London, 36, 325–337.
- 50. Tarantola A., Mullis J., Vennemann T., Dubessy J., de Capitani C. (2007). Oxidation of methane at the
CH₄/H₂O-(CO₂) transition zone in the external part of the Central Alps, Switzerland: Evidence from stable
isotope investigations. *Chemical Geology*, 237(3), 329–357.
- 51. Tarantola A., Mullis J., Guillaume D., Dubessy J., de Capitani C. and Abdelmoula M. (2009). Oxidation of
CH₄ to CO₂ and H₂O by chloritization of detrital biotite at 270 ± 5 °C in the external part of the Central Alps,
Switzerland. *Lithos*, 112(3), 497–510.
- 52. Thompson, J., Meadows, N.S., Trueblood, S.P., Hardman, M., Cowan, G. (1997). Clastic sabkhas and
diachroneity at the top of the Sherwood Sandstone Group: East Irish Sea Basin. *Petroleum Geology of the*
*Irish Sea and Adjacent Areas*, 124(1), 237–251.
- 53. Tian, H., Xiao, X., Wilkins, R. W., Tang, Y. (2012). An experimental comparison of gas generation from three
oil fractions: Implications for the chemical and stable carbon isotopic signatures of oil cracking gas. *Organic*
*Geochemistry*, 46, 96–112.
- 54. Wenger, L. M., Isaksen, G. H. (2002). Control of hydrocarbon seepage intensity on level of biodegradation in
sea bottom sediments. *Organic Geochemistry*, 33(12), 1277–1292.
- 55. Walanda, D.K., Lawrance, G.A., Donne, S.W. (2005). Hydrothermal MnO₂: synthesis, structure, morphology
and discharge performance. *Journal of Power Sources*, 139(1), 325–341.
- 56. Wolela, A.M., Gierłowski-Kordesch, E.H. (2007). Diagenetic history of fluvial and lacustrine sandstones of
the Hartford Basin (Triassic-Jurassic), Newark Supergroup, USA. *Sedimentary Geology*, 197(1–2), 99–126.
- 57. Worden, R. H., Smalley, P. C. (1996). H₂S-producing reactions in deep carbonate gas reservoirs: Khuff
Formation, Abu Dhabi. *Chemical Geology*, 133(1–4), 157–171.
- 58. Worden, R.H., Smalley, P.C., Oxtoby, N.H. (1996). The effects of thermochemical sulfate reduction upon
formation water salinity and oxygen isotopes in carbonate gas reservoirs. *Geochimica et Cosmochimica Acta*,
60(20), 3925–3931.
- 59. Worden, R.H., Smalley, P.C., Cross, M.M. (2000). The influence of rock fabric and mineralogy on
thermochemical sulfate reduction: Khuff Formation, Abu Dhabi. *Journal of Sedimentary Research*, 70(5),
1210–1221.
- 60. Wu, Y. Y., Ping, J. B., Lu, X. X. (2002). Quantitative research on preservation and destruction of hydrocarbon
accumulation in the northwest edge of Junggar Basin. *Acta Petrolei Sinica*, 23(6), 24–28.

-
- 61. Zheng, G., Xu, W., Etiope, G., Ma, X., Liang, S., Fan, Q., Sajjad, W., Li, Y. (2018). Hydrocarbon seeps in
petroliferous basins in China: A first inventory. *Journal of Asian Earth Sciences*, 151, 269–284.
- 62. Zheng, G., Ma, X., Guo, Z., Hilton, D. R., Xu, W., Liang, S., Fan, Q., Chen, W. (2017). Gas geochemistry and
methane emission from Dushanzi mud volcanoes in the southern Junggar Basin, NW China. *Journal of Asian
Earth Sciences*, 149, 184–190.
- 63. Zheng, J. P., Wang, F.Z., Cheng, Z.M., Wu, X.Z., Zhang, Y.J. (2000). Nature and evolution of amalgamated
basement of Junggar Basin , northwestern China: Sr-Nd isotope evidences of basement igneous rock. *Earth
Science-Journal of China University of Geosciences*, 25(2), 179–185 (In Chinese with English abstract).

Reviewers' comments:

Reviewer #1 (Remarks to the Author):

This is the second time I read the manuscript of Wen-Xuan Hu et al. "Thermochemical oxidation of methane induced by high-valence metal oxides in the deep sedimentary basins".

I am convinced by the scientific replies to the many comments and remarks and to the effort put in increasing the quality of the figures. However the English language (grammar, orthograph) of the replies (blue text) is relatively poor and needs to be checked in detail by a native English speaker (even the title needs to be reconsidered from the English style point of view).

Nancy, January 18th 2018,

A. Tarantola

Reviewer #4 (Remarks to the Author):

The MS is very interesting and presented materials to partially support its conclusions. However, this MS did not consider other possibilities properly. Thus, a major revision is required. My comments are as follow:

1) Lines 34-35: TSR by methane was first proposed by Worden et al. (1996), however, Machel (2001) argued against it. Krouse et al. (1998) did not mention any methane oxidization during TSR. Some pioneer works such as Cai et al. (2003; 2004) in this area should be cited. So far very few carbonate cements have been found to have typical feature of TSR by methane. The authors should re-read through those references carefully and revise it correspondingly.

2) Calcite cement distribution and crystalline shape are used to distinguish early and late calcite in this MS, however, they are not sufficient. Fluid inclusion homogenization temperatures and cathode luminescence are widely used to help to solve this issue.

3) Assuming that early and late calcite are properly defined, differences in Mn content of both phases are not much in Fig. 2, and thus the relationship between MnO and calcite stage is not clear. In fact, some samples have much lower MnO content (0.79%) should be measured for carbon isotopes. If the proposal in this text is right, all MnO in this sample should be from early calcite, its carbon isotopic composition should be serve as an end member of early calcite? The other question is, if late calcite is from TOM, how about early calcite? from AOM?

4) Fig. 6 is interesting. However, more discussion is required. Generally C2-C4 is preferentially oxidized compared with C1, but not supported by Fig. 6. It is quite possible that CO₂ cannot be used to reflect oxidization extent in this case. CO₂/(CO₂+C1-4) can be used only when CO₂ show negative shift in carbon isotopes with increasing CO₂/(CO₂+C1-4). CO₂ carbon isotopes should be present.

5) More data should be present to support the assumption that the solution to precipitate late calcite to have Oxygen isotopic composition from -4 to -6 per mil. Because we just know that these conglomerates and sandstones were deposited from dry and meteoric water with an original value from -10 to -8 per mil in this area. The value shall be heavier with enhancing diagenesis. If the low value of -8 per mil is used, most of the calcite shall have much lower precipitation temperatures than in the text. Thus, why the -4 to -6 per mil is used shall be discussed.

6) Carbon isotope fractionation factor acquired from the experiments at significantly higher temperatures than nature should be used with caution. So far, no fractionation factor of TSR by alkanes from the simulation has been used for natural system despite of a dozen of such experiments, let alone Mn oxidization. Two TSR cases show similar carbon isotope fractionation should be cited.

7) Much ¹²C-rich C in calcite than in methane may have been derived from, a) earlier CH₄ charge when only very small part of methane was cracked from kerogen, or, b) from biological methane. More discussion is required.

8) Methane must have been oxidized in this case. However, the MS does not explain why methane was oxidized rather than other alkanes with much lower Gibbs free energy increment (ΔG). It is generally accepted that methane is the last alkane to be oxidized (Machel, 2001).

9) Some much-related references should be added and some not much related should be removed.

Detailed Response to Reviewers

Itemized and detailed responses ('R') to the reviewers' comments are appended below. Key revisions have been highlighted in blue in the revised manuscript.

1. Response to Reviewer 1# (Dr. Alexandre Tarantola)

(1) The English language (grammar, orthograph) of the replies (blue text) is relatively poor and needs to be checked in detail by a native English speaker (even the title needs to be reconsidered from the English style point of view).

R: Thanks a lot for your careful review and constructive comments. Following your advice, we turned to an American friend major in Geochemistry (Dr. Shane Schoepfer, assistant professor at Western Carolina University) for improving the language quality of the revised text, especially the introduction section. Through editing, the current manuscript language is well-written overall. The replies in the last revised manuscript (Line 1 on Page 2 to Line 6 on Page 3, Lines 9–12 on Page 4, Lines 11–15 on Page 12, and Lines 8–15 on Page 18) were still highlighted in blue, which can be checked. We also modified the title for a more accurate expression.

2. Response to Reviewer 4# (Anonymous Reviewer)

(1) Lines 34-35: TSR by methane was first proposed by Worden et al. (1996), however, Machel (2001) argued against it. Krouse et al. (1988) did not mention any methane oxidization during TSR. Some pioneer works such as Cai et al. (2003; 2004) in this area should be cited. So far very few carbonate cements have been found to have typical feature of TSR by methane. The authors should re-read through those references carefully and revise it correspondingly.

R: Thanks a lot for your insightful comments. As you suggested, we re-read related references

carefully. You are correct that, so far, few carbonate cements have been found to have features
typical of TSR by methane. Methane is generally believed to be the most stable of the light
hydrocarbon gases and the last one to undergo TSR, due to a relatively high Gibbs free energy of
reaction (Machel, 2001; Cai et al., 2003, 2004; Hao et al., 2008; Liu et al, 2013). Krouse et al. (1988)
highlighted the oxidation of light hydrocarbon gases, especially ethane and propane. However,
methane can be the predominant organic reactant for TSR when the natural gas reaches high dryness
coefficient values ($C_1/\Sigma C_{1-6}$) (Cai et al., 2004, 2013; Hao et al., 2008; Liu et al., 2013). Methane
was reported to be an effective reductant during TSR in the Permian Khuff Formation of Abu Dhabi
and Saudi Arabia (Worden and Smalley, 1996; Worden et al., 1996; Jenden et al., 2015), and the
Triassic Jialingjiang and Feixianguan formations in the East Sichuan Basin, China (Cai et al., 2003,
2004). Relevant citations were revised accordingly (Lines 2–4 on Page 3) and more discussion was
added to the revised manuscript (Lines 12–14 on Page 10 and Lines 18–19 on Page 13).

(2) Calcite cement distribution and crystalline shape are used to distinguish early and late calcite in
this MS, however, they are not sufficient. Fluid inclusion homogenization temperatures and cathode
luminescence are widely used to help to solve this issue.

**R:** Thank you for your suggestions. At the start of this study, we carried out detailed
petrological and mineralogical optical microscopy on 103 conglomerate and sandstone thin sections
half-stained by Alizarin Red S. To determine the stages (or generations) of authigenic calcites, 38
polished thin sections were analyzed via backscattered electron imaging (BSE) using an electron
probe micro-analyzer (EPMA). Following your advice, another 20 polished thin sections were
observed by cold-cathode CL. Results of the optical microscopy show that calcites mainly occur as
interparticle cements, although small quantities fill the dissolution pores in feldspars (Fig. R1a).

On cold-cathode CL pictures, the calcites are orange and dark red, corresponding to early-stage
 and late-stage calcites respectively (Fig. R1b, c). The former occur as xenomorphic crystals filling
 dissolved pores in feldspars or poorly-connected interparticle pores, whereas the late-stage cements
 appear as coarse crystals, with well-developed crystal planes in the central portions of large
 interparticle pores. The two stages of calcite usually occur together (Fig. R1b, c). However, the
 quality of CL pictures was easily degraded by the flatness of the thin sections and exposure time
 during imaging; unfortunately a large proportion of these pictures were poor-quality and not are,
 therefore, presented in the manuscript.

**Fig. R1.** Typical images of authigenic calcites in T_1b from: optical microscopy (a), cold-cathode CL
(b) and (c), and BSE imaging (d)–(f). (a) in cross-polarized light, calcites occur as interparticle
cements, and fill the dissolution pores of microcline with crosshatched crystal twinning (section from
T_1b_1 at 3819.0 m in Well M602); (b) under the cold-cathode CL, the early-stage calcite appears
orange and some anomalous yellow spots occur on the calcite surface (section from T_1b_1 at 3819.0 m
in Well M602); (c) the early-stage calcite is orange, while the late-stage is dark red under the
cold-cathode CL (section from T_1b_2 at 3296.3 m in Well MH4); (d) in BSE pictures, the early-stage
calcite is dark grey, while the late-stage is light grey (section from T_1b_2 at 3296.3 m in Well MH4);
(e) the early-stage calcite occurs as xenomorphic crystals filling dissolved pores in feldspars or
poorly-connected interparticle pores, whereas the late-stage occur as cements (section from T_1b_2 at
3865.3 m in Well M18); (f) the late-stage cements appear as coarse crystals, with well-developed
crystal planes in the central portions of large interparticle pores, (section from T_1b_2 at 3904.9 m in
Well M18). A substantial difference in manganese content of the two stages calcites is apparent in (d)
to (f).

Referring to the back-scattered electron (BSE) images, two stages of calcite exhibit obviously
different brightnesses, i.e., the early-stage calcite is dark grey, while the late-stage is light grey (Fig.
R1d-f). Compared to the cold-cathode CL pictures, the BSE imaging was less-affected by the
flatness of the thin sections, as they were coated with a carbon film that allowed the calcite stages to
be seen more clearly. During BSE imaging, the major elements of authigenic calcites were analyzed
on EPMA. We found that the manganese contents are remarkably different in the two stages of
calcite (Fig. R1d-f). In detail, the MnO content of early-stage calcite cements is generally lower than

4.00 wt.%, while late-stage cements typically have MnO content higher than 5.00 wt.%, with some
samples showing values as high as 11.00 – 15.00 wt.% (Fig. R1d-f).

Furthermore, to precisely constrain the stages of calcites as you advised, the homogenization
temperatures (T_h) of primary two-phase aqueous fluid inclusions were measured. However, it was
challenging to find fluid inclusions in the calcite cements from the study area. The microscopic view
was very dim because of the strong birefringence of calcite and the small size of the fluid inclusions,
most of which were less than 5 μm long (Fig. R2). We were able to obtain 15 valid values from 32
double-side polished sections. The T_h values are 92, 93, 97, 106, 107, 107, 112, 113, 116, 118, 122,
123, 126, 128, and 132 $^{\circ}\text{C}$, effectively falling in two intervals of 90 – 100 $^{\circ}\text{C}$ (early-stage calcites)
and 105 – 135 $^{\circ}\text{C}$ (late-stage calcites). Since the number of data points is so limited, we ultimately
decided not to show them in the revised manuscript.

**Fig. R2.** The occurrence of two-phase aqueous inclusions and their homogenization temperatures in
authigenic calcites of T_1b .

(3) Assuming that early and late calcite are properly defined, differences in Mn content of both
phases are not much in Fig. 2, and thus the relationship between MnO and calcite stage is not clear.
In fact, some samples have much lower MnO content (0.79%) should be measured for carbon
isotopes. If the proposal in this text is right, all MnO in this sample should be from early calcite, its

carbon isotopic composition should be serve as an end member of early calcite? The other question is,
if late calcite is from TOM, how about early calcite? from AOM?

**R:** Thanks for your comments. The early-stage calcite MnO value reaches 4.32 wt. % in Point
3# (P3) of Fig.2a. This point is indeed a typical one. However, in this sample all of the calcites filling
the feldspar dissolution pores are indeed early-stage based on BSE and CL observations (Fig.2a; Fig.
R3). As mentioned in the manuscript, the manganese content in the calcites of two stages is quite
different, but it does not mean the two stages can only be separated by their manganese content.

**Fig. R3.** Microscope (a) and CL (b) pictures of the calcites filling the feldspar dissolution pores in
the sandy gravelly conglomerate of T₁b₂ at 3824.2 m in Well M18.

Regarding the question about the low MnO content calcite, although the two-stage calcites
occur together in most samples, we identified two samples (M18-14 and AH1-8) that contain only
early-stage calcites. All of the values of MnO content are very low (< 2.50 wt. %) (Fig. R4). We
agree with you that in these two samples the carbon isotopic composition could serve as an end
member of early-stage calcite. The $\delta^{13}\text{C}$ values are -41.7 ‰ and -39.1 ‰ for M18-14 and AH1-8,
respectively. Compared to the $\delta^{13}\text{C}$ values of the T₁b samples, these values are relatively high (Fig.
4a in the manuscript), higher than the typical value of AOM-related authigenic carbonates (-125 ‰
to -45 ‰ , e.g. Ritger et al., 1987; Campbell et al., 2002; Peckmann and Thiel, 2004; Drake et al.,
2015). AOM-related calcites often have much lower $\delta^{13}\text{C}$ values, due to highly ^{13}C -depleted biogenic

methane and the microbial isotope fractionations that occur during methane oxidation (Whiticar,
1999; Machel et al., 2001; Drake et al., 2015); whereas, the $\delta^{13}\text{C}$ value of TOM-related calcites is
relatively higher. Hence, we infer that the early-stage calcites in T_{1b} are also derived from the
thermochemical oxidation of hydrocarbons, especially methane. In comparison, the $\delta^{13}\text{C}$ values of
late-stage calcites are lower.

**Fig. R4.** BSE images of samples of T_{1b} only containing early-stage calcite. (a) Sample *M18-14* with
a bulk calcite $\delta^{13}\text{C}$ value of -41.7‰ from 3914.8 m in T_{1b_1} of Well M18; (b) Sample *AH1-8* with a
bulk calcite $\delta^{13}\text{C}$ value of -39.1‰ , from 3814.5 m in T_{1b_2} of Well AH1.

(4) Fig. 6 is interesting. However, more discussion is required. Generally $\text{C}_2\text{-C}_4$ is preferentially
oxidized compared with C_1 , but not supported by Fig. 6. It is quite possible that CO_2 cannot be used
to reflect oxidization extent in this case. $\text{CO}_2/(\text{CO}_2+\text{C}_{1-4})$ can be used only when CO_2 show negative
shift in carbon isotopes with increasing $\text{CO}_2/(\text{CO}_2+\text{C}_{1-4})$. CO_2 carbon isotopes should be present.

**R:** Nice comments. We carried out stable carbon isotope analyses on CO_2 together with $\text{C}_1\text{-C}_4$
on CG-IRMS one-and-a-half years ago. Unfortunately, valid values of CO_2 were not obtained during
double test attempts because of the low CO_2 content in the gas samples. Vast literatures have verified
that $\text{C}_2\text{-C}_4$ are preferentially involved in TSR reactions when the dryness of the gas is not too high
(e.g., Krouse et al., 1988; Machel, 2001; Cai et al., 2003, 2004; Hao et al., 2008; Liu et al., 2013).

Your observations inspired us to think deeply on why preferential oxidation of C₂–C₄ was not
detected in the T_{1b} clastic rocks.

Taking the geological setting into consideration, the thermochemical oxidation in our study area
that is dominated by methane should be closely related to the occurrence of oxidizing agents (i.e.
Mn^{3+/4+} and small quantities of Fe³⁺ in hematite, Fig. S4). Hematite is widespread in the clay matrix
of brown conglomerates and sandstones, as well as in the brown mudstones in T_{1b}. As shown in Fig.
S4, hematite occurs as isolated amorphous aggregates in the matrix, with an average Mn₂O₃ content
of 1.14 wt.%, or is disseminated in the clay with an average Mn₂O₃ content of 0.68 wt.%. In weakly
acidic fluids, Mn^{3+/4+} is released into the formation water through the dissolution of hematite
(Walanda et al., 2005; Artamonova et al., 2013). In TSR geological systems, the oxidizing agent
(aqueous SO₄²⁻) originally occurs as nodular anhydrite in carbonate rocks or thin anhydrite
interlayers within carbonate deposits (Worden et al., 1996; Cai et al., 2003, 2004; Jenden et al., 2015).
TSR reactions occur in the gas zone or gas-water transition zone where the supply of light
hydrocarbon gases (C₁–C₄) is sufficient (Krouse et al., 1988; Worden et al., 1996; Machel, 2001; Cai
et al., 2003, 2004). However, the thermochemical oxidation of hydrocarbons in T_{1b} clastic rocks
occurred throughout the strata, wherever the gas could reach through diffusion, and was not limited
to the specific gas zone or gas-water transition zone (Krouse et al., 1988; Machel, 2001; Cai et al.,
2003, 2004). During the reactions, gaseous alkanes reduced the high-valence Mn/Fe in hematite in
the clay matrix of reservoir rocks or brown mudstones and “bleached” these clastic rocks. Similar
“bleaching” processes were also reported in some red beds, such as the Jurassic sandstones on the
Colorado Plateau of southern Utah (Beitler et al., 2003; Chan et al., 2000, 2004; Parry et al., 2004,
2009; Wigley et al., 2012), and the Permian Rush Springs Formation red sandstones in the Cement

field of Oklahoma (Elmore and Leach, 1990; Sun and Khan, 2016). Although methane is the last
alkane to be oxidized because of its higher Gibbs free energy increment compared to C₂–C₄ (Machel,
2001; Cai et al., 2003, 2004), the supply of hematite within brown clastic rocks is sufficient for
almost all of the gaseous alkanes, including methane, to be consumed as they move along the
migration pathways. Ultimately, as the main composition of the T_{1b} natural gas, methane dominated
the thermochemical oxidation that was mainly induced by Mn^{3+/4+} in hematite and consequently, the
majority of the calcite δ¹³C values are even lower than those of methane (Fig. 5).

On the other hand, as the clay matrix of reservoir rocks and mudstone are poorly permeable,
having narrow pore throats with diameters of 5 to 100 nm, gaseous alkanes migrated mainly by
diffusion (Nelson, 2009; Peng et al., 2012). Diffusion of gas in clastic rocks depends on its
concentration and the diffusion coefficient shown as

$$Q_t = \frac{DC_0}{l} \left(t - \frac{l^2}{6D} \right)$$

In the formula, Q_t is the total quantity of diffusing gas, D is the gas diffusion coefficient in water, C_0
is the initial gas concentration at the oil-gas interface, l is the distance from the oil-gas interface and t
is time (Crank, 1975). The initial concentration of methane is far higher than that of C₂–C₄ at the
oil-gas interface, in view of the natural gas composition of T_{1b}. Furthermore, the diffusion coefficient
of methane is higher than that of C₂–C₄ (Witherspoon and Saraf, 1965; Matthews and Spearing,
1992). Ultimately, the quantity of methane reaching the matrix of reservoir rocks or mudstones at a
distance from the oil-gas interface is much higher than that of C₂–C₄, which maybe an important
factor resulting in the significant oxidation of methane in the study area.

Following your suggestion, a new paragraph was added in the revised manuscript (Line 18 on
Page 13 to Line 4 on Page 14).

(5) More data should be present to support the assumption that the solution to precipitate late calcite
to have Oxygen isotopic composition from -4 to -6 per mil. Because we just know that these
conglomerates and sandstones were deposited from dry and meteoric water with an original value
from -10 to -8 per mil in this area. The value shall be heavier with enhancing diagenesis. If the low
value of -8 per mil is used, most of the calcite shall have much lower precipitation temperatures than
in the text. Thus, why the -4 to -6 per mil is used shall be discussed.

**R:** Thank you for your comments. As soon as we received them, we tried to collect several
oilfield water samples from T_{1b} to measure their oxygen isotopic composition. The chief geologist
Dr. Xu-Long Wang responsible for the exploration of the Mahu Sag and the senior engineer Mr.
Jian-Guo Wang in the Aihu Oilfield were consulted on the water samples. To our disappointment, no
water has yet been produced from 32 development wells drilled into the T_{1b} reservoir.

However, in the revised manuscript the reason for using the formation-water $\delta^{18}\text{O}$ value of
13 -6‰ to -4‰ was explained in detail (Lines 10–14 on Page 11). As you mentioned, the formation
waters become progressively enriched in ^{18}O in clastic strata with enhancing interactions between
silicate minerals and water, so their $\delta^{18}\text{O}$ values are closely related to the diagenetic stage of the rock
(Egeberg and Aagaard, 1989; Haszeldine et al., 1992; Giles et al., 1992; Marchand et al., 2002;
Wilkinson et al., 2004; Harwood et al., 2013). Based on systematic studies of diagenesis and oxygen
isotopes of authigenic minerals, a reliable reference on relationships between pore water evolution
and diagenetic processes has been established in the well-known Brent Group sandstone (Middle
Jurassic) of the North Sea by Bjørlykke et al. (1992), Giles et al. (1992), Haszeldine et al. (1992),
Wilkinson et al. (2004) and others. They reported that in this group, at burial depths of 2500–3500 m,
albitization of K-feldspar, dissolution of K-feldspar, and the precipitation of kaolinite, authigenic
carbonates, as well as small quantities of quartz occur; however, illitization of kaolinite is not
observed. At this diagenetic stage, the $\delta^{18}\text{O}$ value of pore water has increased by at least 4 ‰, from

1 -7 ‰ of the initial meteoric water to -3 ‰ (Haszeldine et al., 1992; Wilkinson et al., 2004).
 Similarly, Zhang et al. (2015) and Kang et al. (2018) report a series of comparable diagenetic
 phenomena, such as albitization of K-feldspar, dissolution of K-feldspar, and precipitation of small
 quantities of quartz, with no illitization of kaolinite, in the Baikouquan Formation. Since T_{1b}
 underwent diagenesis in a closed diffusive system, its pore water δ¹⁸O can be expected to have
 increased by a quantity similar to that of the Brent Group (about 4 ‰) to -6 to -4 ‰, attempting to
 reach equilibrium with the rock from the δ¹⁸O of the initial meteoric water (-10 to -8 ‰) (Sun et al.,
 2009; Roberts et al., 2018). Therefore, we used a δ¹⁸O range of -6 ‰ to -4 ‰ to calculate the
 precipitation temperature of calcites. Results indicate that the precipitation of Mn-rich calcite
 occurred at 90 – 135 °C (Fig. R5). These values are consistent with the homogenization temperatures
 of two-phase aqueous inclusions in calcite cements.

**Fig. R5.** Calculated precipitation temperature of authigenic calcites in the T_{1b} reservoir rocks.

(6) Carbon isotope fractionation factor acquired from the experiments at significantly higher
 temperatures than nature should be used with caution. So far, no fractionation factor of TSR by
 alkanes from the simulation has been used for natural system despite of a dozen of such experiments,
 let alone Mn oxidization. Two TSR cases show similar carbon isotope fractionation should be cited.

**R:** Thanks for your constructive comments. Although there is only experimental data on the
 carbon isotope fractionation factor α in the oxidation of methane by high-valence Mn oxides, the data
 of similar geological reactions is also a valuable reference, as you advised. The relevant literature on

TSR was investigated. Cai et al. (2013) calculated the factor α during methane-dominated TSR
through original work in the East Sichuan Basin of China, and they reported a value of 1.0166,
indicating a relatively high reaction rate. Furthermore, Mankiewicz et al. (2009) and Cai et al. (2013)
reported a value of 1.109 as α for ethane undergoing TSR in the Mobile Bay Jurassic Norphlet
Formation. According to the empirical equation obtained from experiments on methane oxidation by
MnO₂ (Kiyosu and Imaizumi, 1996), we calculate an α of 1.0175 – 1.0193 for a closed system at 90 –
135 °C, which is close to the α of the two TSR cases above. Considering the similarity of the
mechanisms involved in the oxidation of methane by high-valence Mn oxides and TSR, the values
were added in the revised manuscript (Lines 4–6 on Page 15).

(7) Much ¹²C-rich C in calcite than in methane may have been derived from, a) earlier CH₄ charge
when only very small part of methane was cracked from kerogen, or, b) from biological methane.
More discussion is required.

**R:** Agree with your comments. During the thermochemical oxidation of methane by
high-valence Mn(Fe) oxides, significantly ¹³C-depleted calcite can be formed from the extreme
kinetic fractionation of carbon isotopes during reactions or an originally ¹³C-depleted methane, such
as biogenic methane (Kiyosu and Krouse, 1989; Kiyosu and Imaizumi, 1996; Pan et al., 2006; Cai et
al., 2013; Drake et al., 2015). As to AOM-related calcites, their much lower $\delta^{13}\text{C}$ values are caused
by the highly ¹³C-depleted biogenic methane and the microbial isotope fractionations during methane
oxidation (Whiticar, 1999; Machel et al., 2001; Campbell et al., 2002; Peckmann and Thiel, 2004;
Drake et al., 2015). The T_{1b} reservoir rocks in the study area were charged by oil and gas from the
lacustrine sapropelic-type Fengcheng Formation (P_{1f}) source rocks during the Early Jurassic and
Early Cretaceous (Cao et al., 2005; Ablimit et al., 2015; Qi et al., 2015; Tao et al., 2016). By the

petroleum accumulation stage, the P_{1f} source rocks had matured to be mature to highly mature for
hydrocarbon generation (Cao et al., 2005; Tao et al., 2016). The reservoir rocks were mainly charged
by the light oil and gaseous products of oil-cracking from the mature to highly mature source rocks
(Lei et al., 2014; Ablimit et al., 2015; Tao et al., 2016), but the possibility that T_{1b} might be charged
by very small quantities of methane derived from kerogen-cracking cannot be totally ruled out.
Related expressions were added in the revised manuscript (Lines 11–13 on Page 15).

However, it is much unlikely that biogenic methane was oxidized in T_{1b} . Firstly; organic matter
is depleted throughout the T_{1b} sedimentary assemblages, as recorded by abundant subsurface brown
mudstones and conglomerates of the formation (Fig. S3). There was no material basis for the
formation of biogenic methane in T_{1b} during the early diagenesis. Secondly; if small volumes of
biogenic methane were formed in the underlying Permian source rocks and only that methane
charged T_{1b} in its early diagenesis, early-stage calcite with extremely low $\delta^{13}C$ values would
precipitate. However, the $\delta^{13}C$ values of early-stage calcite samples typical of T_{1b} are relatively high
(see our response to your 3rd question); thus, the possibility of biogenic methane oxidation can be
ruled out.

(8) Methane must have been oxidized in this case. However, the MS does not explain why methane
was oxidized rather than other alkanes with much low the Gibbs free energy increment (ΔG). It is
generally accepted that methane is the last alkane to be oxidized (Machel, 2001).

**R:** Thanks for your constructive comments. As you mentioned, methane is generally believed to
be the most stable of the light hydrocarbon gases and the last one to undergo TSR (Krouse et al.,
1988; Machel, 2001). However, methane can also be the predominant organic reactant for TSR,
when the natural gas has a high dryness coefficient (Cai et al., 2004, 2013; Hao et al., 2008; Liu et al.,

2013). The debate about why abundant methane was oxidized in T_{1b} clastic strata was explained in
detail in the reply to your 4th question.

To better address this issue, a new paragraph was added to the revised manuscript Line 18 on
Page 13 to Line 4 on Page 14).

(9) Some much-related references should be added and some not much related should be removed.

**R:** Thank you for this valid point. After revision, the number of citations has been reduced from
69 to 61. Please see the Reference section in the revised manuscript.

References

1. Abulimiti, Cao, J., Chen, J., Yang, H., Chen, G., Tao, K. (2015). Origin and occurrence of highly matured oil and gas in Mahu sag, Junggar Basin. *Xinjiang Petroleum Geology*, 36(4), 379–384 (In Chinese with English abstract).
2. Artamonova, I. V., Gorichev, I. G., Godunov, E. B. (2013). Kinetics of manganese oxides dissolution in sulphuric acid solutions containing oxalic acid. *Engineering*, 5(09), 714–719.
3. Beitler, B., Chan, M. A., Parry, W. T. (2003). Bleaching of Jurassic Navajo sandstone on Colorado Plateau Laramide highs: Evidence of exhumed hydrocarbon supergiants?. *Geology*, 31(12), 1041–1044.
4. Bjørlykke, K., Nedkvitne, T., Ramm, M., Saigal, G. C. (1992). Diagenetic processes in the Brent Group (Middle Jurassic) reservoirs of the North Sea: an overview. *Geological Society, London, Special Publications*, 61(1), 263–287.
5. Cai, C., Worden, R. H., Bottrell, S. H., Wang, L., Yang, C. (2003). Thermochemical sulphate reduction and the generation of hydrogen sulphide and thiols (mercaptans) in Triassic carbonate reservoirs from the Sichuan Basin, China. *Chemical Geology*, 202(1–2), 39–57.
6. Cai, C., Xie, Z., Worden, R. H., Hu, G., Wang, L., He, H. (2004). Methane-dominated thermochemical sulphate reduction in the Triassic Feixianguan Formation East Sichuan Basin, China: towards prediction of fatal H₂S concentrations. *Marine and Petroleum Geology*, 21(10), 1265–1279.
7. Cai, C., Zhang, C., He, H., Tang, Y. (2013). Carbon isotope fractionation during methane-dominated TSR in East Sichuan Basin gasfields, China: A review. *Marine and Petroleum Geology*, 48, 100–110.
8. Campbell, K. A., Farmer, J. D., Des Marais, D. (2002). Ancient hydrocarbon seeps from the Mesozoic convergent margin of California: carbonate geochemistry, fluids and palaeoenvironments. *Geofluids*, 2(2), 63–94.
9. Cao, J., Zhang, Y., Hu, W., Yao, S., Wang, X., Zhang, Y., Tang, Y. (2005). The Permian hybrid petroleum system in the northwest margin of the Junggar Basin, northwest China. *Marine and Petroleum Geology*, 22, 331–349.
10. Chan, M. A., Beitler, B., Parry, W. T., Ormö, J., Komatsu, G. (2004). A possible terrestrial analogue for haematite concretions on Mars. *Nature*, 429(6993), 731–734.
11. Chan, M. A., Parry, W. T., Bowman, J. R. (2000). Diagenetic hematite and manganese oxides and fault-related fluid flow in Jurassic sandstones, southeastern Utah. *AAPG Bulletin*, 84(9), 1281–1310.
12. Crank, J. (1975). *The Mathematics of Diffusion*. Clarendon Press, 44–69.
13. Drake, H., Aström, M. E., Heim, C., Broman, C., Aström, J., Whitehouse, M., ... Sjövall, P. (2015). Extreme ¹³C depletion of carbonates formed during oxidation of biogenic methane in fractured granite. *Nature Communications*, 6, 7020.
14. Egeberg, P. K., Aagaard, P. (1989). Origin and evolution of formation waters from oil fields on the Norwegian shelf. *Applied Geochemistry*, 4(2), 131–142.

- 15. Elmore, R. D., Leach, M. C. (1990). Remagnetization of the Rush Springs Formation, Cement, Oklahoma:
Implications for dating hydrocarbon migration and aeromagnetic exploration. *Geology*, 18(2), 124–127.
- 16. Giles, M. R., Stevenson, S., Martin, S. V., Cannon, S. J. C., Hamilton, P. J., Marshall, J. D., Samways, G. M.
(1992). The reservoir properties and diagenesis of the Brent Group: a regional perspective. *Geological Society,
London, Special Publications*, 61(1), 289–327.
- 17. Hao, F., Guo, T., Zhu, Y., Cai, X., Zou, H., Li, P. (2008). Evidence for multiple stages of oil cracking and
thermochemical sulfate reduction in the Puguang gas field, Sichuan Basin, China. *AAPG Bulletin*, 92(5),
611–637.
- 18. Harwood, J., Aplin, A. C., Fialips, C. I., Iliffe, J. E., Kozdon, R., Ushikubo, T., Valley, J. W. (2013). Quartz
cementation history of sandstones revealed by high-resolution SIMS oxygen isotope analysis. *Journal of
Sedimentary Research*, 83(7), 522–530.
- 19. Haszeldine, R. S., Brint, J. F., Fallick, A. E., Hamilton, P. J., Brown, S. (1992). Open and restricted
hydrologies in Brent Group diagenesis: North Sea. *Geological Society, London, Special Publications*, 61(1),
401–419.
- 20. Jenden, P. D., Titley, P. A., Worden, R. H. (2015). Enrichment of nitrogen and ¹³C of methane in natural gases
from the Khuff Formation, Saudi Arabia, caused by thermochemical sulfate reduction. *Organic Geochemistry*,
82, 54–68.
- 21. Kang, X., Hu, W., Cao, J., Jin, J., Wu, H., Zhao, Y., Wang, J. (2018). Selective dissolution of alkali feldspars
and its effect on Lower Triassic sandy conglomerate reservoirs in the Junggar Basin, northwestern China.
*Geological Journal*, 53(2), 475–499.
- 22. Kiyosu, Y., Imaizumi, S. (1996). Carbon and hydrogen isotope fractionation during oxidation of methane by
metal oxides at temperatures from 400 to 530 C. *Chemical Geology*, 133(1–4), 279–287.
- 23. Kiyosu, Y., Krouse, H. R. (1989). Carbon isotope effect during abiogenic oxidation of methane. *Earth and
Planetary Science Letters*, 95(3–4), 302–306.
- 24. Krouse, H.R., Viau, C.A., Eliuk, L.S., Ueda, A., Halas, S. (1988). Chemical and isotopic evidence of
thermochemical sulphate reduction by light hydrocarbon gases in deep carbonate reservoirs. *Nature*,
333(6172), 415–419.
- 25. Lei, D., Abulimiti, Tang, Y., Chen, J., Cao, J. (2014). Controlling Factors and Occurrence Prediction of High
Oil-Gas Production Zones in Lower Triassic Baikouquan Formation of Mahu Sag in Junggar Basin. *Xinjiang
Petroleum Geology*, 35(5), 495–499.
- 26. Liu, Q.Y., Worden, R.H., Jin, Z.J., Liu, W.H., Li, J., Gao, B., Zhang, D.W., Hu, A.P., Yang, C. (2013). TSR
versus non-TSR processes and their impact on gas geochemistry and carbon stable isotopes in Carboniferous,
Permian and Lower Triassic marine carbonate gas reservoirs in the Eastern Sichuan Basin, China. *Geochimica
et Cosmochimica Acta*, 100, 96–115.
- 27. Machel, H. G. (2001). Bacterial and thermochemical sulfate reduction in diagenetic settings—old and new
insights. *Sedimentary Geology*, 140(1), 143–175.
- 28. Mankiewicz, P. J., Pottorf, R. J., Kozar, M. G., Vrolijk, P. (2009). Gas geochemistry of the Mobile Bay
Jurassic Norphlet Formation: Thermal controls and implications for reservoir connectivity. *AAPG Bulletin*,
93(10), 1319–1346.

- 29. Marchand, A. M., Macaulay, C. I., Haszeldine, R. S., Fallick, A. E. (2002). Pore water evolution in oilfield
sandstones: constraints from oxygen isotope microanalyses of quartz cement. *Chemical Geology*, 191(4),
285–304.
- 30. Matthews, G. P., Spearing, M. C. (1992). Measurement and modelling of diffusion, porosity and other pore
level characteristics of sandstones. *Marine and petroleum geology*, 9(2), 146–154.
- 31. Nelson, P. H. (2009). Pore-throat sizes in sandstones, tight sandstones, and shales. *AAPG Bulletin*, 93(3),
329–340.
- 32. Pan, C., Yu, L., Liu, J., Fu, J. (2006). Chemical and carbon isotopic fractionations of gaseous hydrocarbons
during abiogenic oxidation. *Earth and Planetary Science Letters*, 246(1–2), 70–89.
- 33. Parry, W. T., Chan, M. A., Beitler, B. (2004). Chemical bleaching indicates episodes of fluid flow in
deformation bands in sandstone. *AAPG Bulletin*, 88(2), 175–191.
- 34. Parry, W. T., Chan, M. A., Nash, B. P. (2009). Diagenetic characteristics of the Jurassic Navajo Sandstone in
the Covenant oil field, central Utah thrust belt. *AAPG Bulletin*, 93(8), 1039–1061.
- 35. Peckmann, J., Thiel, V. (2004). Carbon cycling at ancient methane-seeps. *Chemical Geology*, 205(3–4),
443–467.
- 36. Peng, S., Hu, Q., Hamamoto, S. (2012). Diffusivity of rocks: Gas diffusion measurements and correlation to
porosity and pore size distribution. *Water Resources Research*, 48(2), 1–9.
- 37. Qi, W., Pan, J., Wang, G., Qu, Y., Tan, K., Yin, L. (2015). Fluid inclusion and hydrocarbon charge history for
reservoir of Baikouquan Formation in the Mahu Sag, Junggar Basin. *Natural Gas Geoscience*, 26, 64–71 (In
Chinese with English abstract).
- 38. Ritger, S., Carson, B., Suess, E. (1987). Methane-derived authigenic carbonates formed by subduction-induced
pore-water expulsion along the Oregon/Washington margin. *Geological Society of America Bulletin*, 98(2),
147–156.
- 39. Roberts, J., Turchyn, A. V., Wignall, P. B., Newton, R. J., Vane, C. H. (2018). Disentangling Diagenesis From
the Rock Record: An Example From the Permo-Triassic Wordie Creek Formation, East Greenland.
*Geochemistry, Geophysics, Geosystems*, 19(1), 99–113.
- 40. Sun, L., Khan, S. (2016). Ground-based hyperspectral remote sensing of hydrocarbon-induced rock alterations
at cement, Oklahoma. *Marine and Petroleum Geology*, 77, 1243–1253.
- 41. Sun, Y., Joachimski, M. M., Wignall, P. B., Yan, C., Chen, Y., Jiang, H., ... Lai, X. (2012). Lethally hot
temperatures during the Early Triassic greenhouse. *Science*, 338(6105), 366–370.
- 42. Tao, K., Cao, J., Wang, Y., Ma, W., Xiang, B., Ren, J., Zhou, N. (2016). Geochemistry and origin of natural
gas in the petroliferous Mahu sag, northwestern Junggar Basin, NW China: Carboniferous marine and Permian
lacustrine gas systems. *Organic Geochemistry*, 100, 62–79.
- 43. Walanda, D. K., Lawrance, G. A., Donne, S. W. (2005). Hydrothermal MnO₂: synthesis, structure,
morphology and discharge performance. *Journal of Power Sources*, 139(1), 325–341.
- 44. Whiticar, M. J. (1999). Carbon and hydrogen isotope systematics of bacterial formation and oxidation of
methane. *Chemical Geology*, 161(1–3), 291–314.

- 45. Wigley, M., Kampman, N., Dubacq, B., Bickle, M. (2012). Fluid-mineral reactions and trace metal
mobilization in an exhumed natural CO₂ reservoir, Green River, Utah. *Geology*, 40(6), 555–558.
- 46. Wilkinson, M., Haszeldine, R. S., Ellam, R. M., Fallick, A. (2004). Hydrocarbon filling history from
diagenetic evidence: Brent Group, UK North Sea. *Marine and Petroleum Geology*, 21(4), 443–455.
- 47. Witherspoon, P. A., Saraf, D. N. (1965). Diffusion of methane, ethane, propane, and n-butane in water from 25
to 43 °C. *The Journal of Physical Chemistry*, 69(11), 3752–3755.
- 48. Worden, R. H., Smalley, P. C. (1996). H₂S-producing reactions in deep carbonate gas reservoirs: Khuff
Formation, Abu Dhabi. *Chemical Geology*, 133(1–4), 157–171.
- 49. Worden, R.H., Smalley, P.C., Oxtoby, N.H. (1996). The effects of thermochemical sulfate reduction upon
formation water salinity and oxygen isotopes in carbonate gas reservoirs. *Geochimica et Cosmochimica Acta*,
60(20), 3925–3931.
- 50. Zhang, S. Huan, J., Lei, Z. (2014). Genetic analysis of the high quality reservoir of Triassic Baikouquan
Formation in Mabei region, Junggar Basin. *Acta Sedimentologica Sinica*, 32(6), 1171–1180 (In Chinese with
English abstract).

Reviewers' comments:

Reviewer #4 (Remarks to the Author):

Comments:

This MS has been significantly improved and deserves to be published in this journal due to the new finding about methane oxidization by high valence Mn(III/IV). However, I am still not convinced about the source of the extreme ^{13}C calcite and the amount of oxidized methane calculated.

- 1) In fact, when C1-C4 $\delta^{13}\text{C}$ data in table 6 are plot on Chung et al. (1988), you can see clearly that the methane is not plot along the line and thus is mixture between thermal cracking and biogenic methane. The biogenic methane is expected to have $\delta^{13}\text{C}$ lighter than -46.8 per mil. Thus the most negative value may have derived from the oxidization of pure biogenic methane. In contrast, the heavier calcite may have C derived from thermal oxidization of hydrocarbon but not limited to thermal CH_4 . Although it is reasonable that kinetic isotope fractionation occurs during the oxidization, it is not wise to deduce the fractionation factor in such a complex system with potential reducers including methane, ethane and liquid HCs. IF there exists fractionation as proposed in the text, residual methane is expected to become heavier rather than lighter than thermal cracking methane.

- 2) If calcite has C from other HCs rather than methane as indicated by its varied isotopic values, it is not reasonable to assume that all carbon in the 2.5% calcite in reservoir rocks was derived from methane. If you can extend these calculations to encompass the entire Mahu Sag (~5000 km²), you mean that the whole Sag was charged with methane and other HCs. It seems to me, it is not right. In general, we can find HCs only in part of the area.
- 3) I do not agree about the statement: limited amount of gas found compared with the huge amount of oil is necessarily due to the oxidization of methane. If burial history in Fig. S5 is right, the T1 reservoir never experienced heating >120°C, and the Lower Permian Fengcheng Fm source rock is guessed not >140°C, and thus

peak gas generation is not expected. Additionally, if huge amount of thermal methane was oxidized, the gas is expected to have much lower dryness (not aridity in LINE 128) coefficient than present values (0.77 to 0.95).

- 4) It seems to me that both biogenic and thermal methane have been oxidized by Mn and Fe, however, other HCs may have been oxidized as well in reservoirs with heavier calcite $\delta^{13}\text{C}$. The inhomogeneous distribution of these different HCs and subsequent oxidization during different periods results in calcite having variable $\delta^{13}\text{C}$. Kinetic fractionation must have occurred but is limited compared with the source $\delta^{13}\text{C}$. The amount of oxidized methane is overestimated in this MS.

Detailed Response to Reviewer

Itemized and detailed responses ('R') to the reviewer's comments are appended below. Key revisions have been highlighted in blue in the revised manuscript.

1. Response to Reviewer 4#

(1) In fact, when C₁-C₄ ¹³C data in table 6 are plot on Chung et al. (1988), you can see clearly that the methane is not plot along the line and thus is mixture between thermal cracking and biogenic methane. The biogenic methane is expected to have ¹³C lighter than -46.8 per mil. Thus the most negative value may have derived from the oxidization of pure biogenic methane. In contrast, the heavier calcite may have C derived from thermal oxidization of hydrocarbon but not limited to thermal CH₄. Although it is reasonable that kinetic isotope fractionation occurs during the oxidization, it is not wise to deduce the fractionation factor in such a complex system with potential reducers including methane, ethane and liquid HCs. If there exists fractionation as proposed in the text, residual methane is expected to become heavier rather than lighter than thermal cracking methane.

R: Thank you very much for your insightful comments. Considering the geological background, it is most likely that microbial methane was generated from P_{1f} source rock during the shallow-burial stage. Primary microbial methane production is readily observable in bogs, swamps, and marshes with the humic Type-III kerogen and high TOC (Rice and Claypool, 1981; Ritter et al., 2015; Colosimo et al., 2016). However, the microbial methane production from sapropelic organic matters is severely restricted by competitive substrates in saline lakes or marine environment with abundant dissolved sulphate (Schoell, 1988; Whiticar, 1986, 1999; Martini et al., 1998; Reeburgh, 2007). In P_{1f} source rocks, the hydrocarbon-generating organic matter is sapropelic Type-II kerogen predominant

(Cao et al., 2005; Tao et al., 2016), and gypsum had been identified in the lacustrine
dolomite-dominant strata (Zhang et al., 2012). Therefore, the accumulation of biogenic methane
might be very limited in *P_{1f}*. Despite of the low content of biogenic methane, we agree with you that
its influence on the isotopic composition of authigenic calcite cannot be ruled out. It is absolutely
possible that primary microbial methane was oxidized by high-valence manganese, though its
quantity involving in the oxidation remains unknown. We tried to investigate and collect
compositional and carbon isotopic data on the natural gas from *T_{1b}* and *P_{1f}* in these days.
Unfortunately, just based on current data we found it impossible to calculate a reliable value on the
quantity of biogenic methane involved in the thermochemical oxidation. We expect to solve this
problem in the future study. However, the discussion related to the involvement of biogenic methane
in TOM was still added to the revised manuscript (Lines 13–17 on Page 16).

For the involvement of hydrocarbons other than CH₄, more discussion had been added in the
last revised manuscript (Line 20 on Page 14 to Line 6 on Page 15).

(2) If calcite has C from other HCs rather than methane as indicated by its varied isotopic values, it is
not reasonable to assume that all carbon in the 2.5% calcite in reservoir rocks was derived from
methane. If you can extend these calculations to encompass the entire Mahu Sag (~5000 km²), you
mean that the whole Sag was charged with methane and other HCs. It seems to me, it is not right. In
general, we can find HCs only in part of the area.

**R:** Thanks a lot for your comments. As stated in our response to your 1st question, we agree
with you that the carbon source of authigenic calcites in *T_{1b}* reservoir rocks is not only restricted to
methane. It's true that the whole sag cannot be fully charged by methane and C₂₊ hydrocarbons.
Following your advice, we reassess the methane consumption in the study area. In the *T_{1b}* clastic

strata, the natural gas-charging area is far larger than that of the liquid hydrocarbon-charging. As
discussed in the manuscript, the volatile light hydrocarbons (i.e. natural gas) are largely involved in
the thermochemical oxidation by $Mn^{3+/4+}$ in hematite. If taking the liquid hydrocarbon-charging area
as the minimum natural gas-charging area, the exhausted methane (C_1) can be calculated based on
the relative content of methane in natural gas. However, ^{13}C -depleted authigenic calcites are not
restricted in the conglomerates and sandstones charged by oil; they can also be found in the wells far
away from the oil-charging area (Fig. R1). As a result, the exact methane consumption should be far
more than C_1 . If extending the distribution area of calcite to the entire Mahu Sag, an overestimated
value (C_2) can be obtained. Therefore, we suggest that it is more suitable to present a methane
consumption value in a range of C_1 to C_2 . Related revisions can be seen in the revised manuscript
(Lines 2–8 on Page 18, and Lines 3–17 on Page 24 in the Supplemental Materials section).

**Fig. R1.** Authigenic calcites of T_1b in the wells far away from oil-charging area in the
plane-polarized light of optical microscopy. (a) calcites occur as interparticle cements in the medium
sandstone. The section stained by alizarin red is from T_1b_3 buried at 2883.2 m in Well Aihu4; (b)
Tissue pores of feldspar in muddy pebbly conglomerate were filled with authigenic calcite. The

1 section was stained by alizarin red. This sample was collected from T_{1b2} which was buried at 4315.3
2 m in Well Mazhong1. The location of Well Aihu4 and Mazhong1 can be seen in Fig. 1 of the
3 manuscript.

(3) I do not agree about the statement: limited amount of gas found compared with the huge amount
of oil is necessarily due to the oxidization of methane. If burial history in Fig. S5 is right, the T1
reservoir never experienced heating >120 °C, and the Lower Permian Fengcheng Fm source rock is
guessed not >140 °C, and thus peak gas generation is not expected. Additionally, if huge amount of
thermal methane was oxidized, the gas is expected to have much lower dryness (not aridity in LINE
128) coefficient than present values (0.77 to 0.95).

**R:** Thanks a lot for your careful review and constructive comments. Based on the construction
of the thermal history of the study area, the maturity of the P_{1f} source rock does not reach the gas
generation peak via oil thermal cracking (Qiu et al., 2001; Cao et al., 2005). The thermogenic
methane is mainly oil-associated gas during the thermocatalytic oil-generating stage. Since its
accumulation is limited, relevant expressions of “limited amount of gas found compared with the
huge amount of oil is necessarily due to the oxidization of methane” is indeed not convincing. We
decide to remove this statement from the text in the revised manuscript (Lines 11–16 on Page 18).

(4) It seems to me that both biogenic and thermal methane have been oxidized by Mn and Fe,
however, other HCs may have been oxidized as well in reservoirs with heavier calcite ¹³C. The
inhomogeneous distribution of these different HCs and subsequent oxidization during different
periods results in calcite having variable ¹³C. Kinetic fractionation must have occurred but is limited
compared with the source ¹³C. The amount of oxidized methane is overestimated in this MS.

**R:** Thank you very much for your constructive comments. More discussions on the possible
mixing of biogenic methane were added in the revised manuscript (Lines 13–17 on Page 16). As
stated in our response to your 2nd question, we also re-estimated the amount of oxidized methane

(Lines 2–8 on Page 18). This item of comment made us paying more attention to the inhomogeneous
distribution of different hydrocarbons and related subsequent oxidization process during different
periods in the future study.

**References**

- 1. Rice, D. D., Claypool, G. E. (1981). Generation, accumulation, and resource potential of biogenic gas. *AAPG*
*Bulletin*, 65(1), 5–25.
- 2. Ritter, D., Vinson, D., Barnhart, E., Akob, D. M., Fields, M. W., Cunningham, A. B., Orem, W., McIntosh, J.
C. (2015). Enhanced microbial coalbed methane generation: a review of research, commercial activity, and
remaining challenges. *International Journal of Coal Geology*, 146, 28–41.
- 3. Colosimo, F., Thomas, R., Lloyd, J. R., Taylor, K. G., Boothman, C., Smith, A. D., Lord, R., Kalin, R. M.
(2016). Biogenic methane in shale gas and coal bed methane: a review of current knowledge and gaps.
*International Journal of Coal Geology*, 165, 106–120.
- 4. Schoell, M. (1988). Multiple origins of methane in the Earth. *Chemical geology*, 71(1–3), 1–10.
- 5. Whiticar, M. J., Faber, E., Schoell, M. (1986). Biogenic methane formation in marine and freshwater
environments: CO₂ reduction vs. acetate fermentation—*isotope evidence*. *Geochimica et Cosmochimica Acta*,
50(5), 693–709.
- 6. Whiticar, M. J. (1999). Carbon and hydrogen isotope systematics of bacterial formation and oxidation of
methane. *Chemical Geology*, 161(1–3), 291–314.
- 7. Martini, A. M., Walter, L. M., Budai, J. M., Ku, T. C. W., Kaiser, C. J., Schoell, M. (1998). Genetic and
temporal relations between formation waters and biogenic methane: Upper Devonian Antrim Shale, Michigan
Basin, USA. *Geochimica et Cosmochimica Acta*, 62(10), 1699–1720.
- 8. Reeburgh, W. S. (2007). Oceanic methane biogeochemistry. *Chemical reviews*, 107(2), 486–513.
- 9. Cao, J., Zhang, Y., Hu, W., Yao, S., Wang, X., Zhang, Y., Tang, Y. (2005). The Permian hybrid petroleum
system in the northwest margin of the Junggar Basin, northwest China. *Marine and Petroleum Geology*, 22,
331–349.
- 10. Tao, K., Cao, J., Wang, Y., Ma, W., Xiang, B., Ren, J., Zhou, N. (2016). Geochemistry and origin of natural
gas in the petroliferous Mahu sag, northwestern Junggar Basin, NW China: Carboniferous marine and Permian
lacustrine gas systems. *Organic Geochemistry*, 100, 62–79.
- 11. Zhang, J., He, Z., Xu, H. B., Ji, C. H., Yuan, Q., Shi, J. A., Lu, X. C. (2012). Petrological characteristics and
origin of permian Fengcheng formation dolomitic rocks in Wuerhe-Fengcheng Area, Junggar Basin. *Acta*
*Sedimentol Sin*, 30(5), 859–867 (In Chinese with English abstract).
- 12. Qiu, N. S., Yang, H. B., Wang, X. L. (2002). Tectono-thermal evolution in the Junggar Basin. *Chinese Journal*
*of Geology*, 37, 423–429 (In Chinese with English abstract).

REVIEWERS' COMMENTS:

Reviewer #1 (Remarks to the Author):

Dear Editors of Nature Communications,

This is the third time I was invited to review the manuscript entitled "Thermochemical oxidation of methane induced by high-valence metal oxides in sedimentary basin" provided by Wen-Xuan Hu et al.

The manuscript is now significantly improved and the responses to reviewers convincing. I have minor editing comments:

Fig. 6: It would certainly be easier to read with a legend showing CH₄ at the bottom, then C₂H₆, C₃H₈ and C₄H₁₀. If you do so, also invert the legend of Fig. 7.

l. 180: Please add references.

l. 185: ",," instead of " ; "

l. 189: remove ",,"

l. 190: ... presence of oxidants...

l. 208: what do you mean by unaltered methane?

l. 212: remove the second "and"

l. 219: "methane oxidation" instead of "methane reduction"

l. 297: the latter

l. 405: 12C and 13C

l. 410: measurable

l. 443, 456: can be estimated as

l. 465: which is roughly

l. 467: 1,224 Tg is not given approximately. To me approximately would be more a range of values.

Supplementary (5) Fig. S4(c) Please indicate why there is a * at Mn₂O₃* in the caption.

l. 146. ...temperatures above 80 °C.

Legend of Table 4: "/" denotes that no analyses were carried out. The precipitation temperatures of calcite were calculated by T =... (Irwin et al., 1977).

With my best regards,

Nancy 09.10.2018

A. Tarantola

Reviewer #1 comments

The manuscript is now significantly improved and the responses to reviewers convincing. I have minor editing comments:

Fig. 6: It would certainly be easier to read with a legend showing CH₄ at the bottom, then C₂H₆, C₃H₈ and C₄H₁₀. If you do so, also invert the legend of Fig. 7.

l. 180: Please add references.

l. 185: “,” instead of “;”

l. 189: remove “;”

l. 190: ... presence of oxidants...

l. 208: what do you mean by unaltered methane?

l. 212: remove the second “and”

l. 219: “methane oxidation” instead of “methane reduction”

l. 297: the latter

l. 405: ¹²C and ¹³C

l. 410: measurable

l. 443, 456: can be estimated as

l. 465: which is roughly

l. 467: 1,224 Tg is not given approximately. To me approximately would be more a range of values.

Supplementary (5) Fig. S4(c) Please indicate why there is a * at Mn₂O₃* in the caption.

l. 146. ...temperatures above 80 °C.

Legend of Table 4: “/” denotes that no analyses were carried out. The precipitation temperatures of calcite were calculated by T =... (Irwin et al., 1977).

Reply: Thank you very much for your careful review. Following your advice, we modified Fig. 6, Supplementary Fig. 4 for easier readability, and revised other 15 language defects one by one in the main manuscript and the supplementary notes.